



# UA-ICON with NWP physics package (version: ua-icon-2.1): mean state and variability of the middle atmosphere

Markus Kunze[1], Christoph Zülicke[1], Tarique A. Siddiqui[1,*], Claudia C. Stephan[1], Yosuke Yamazaki[1], Claudia Stolle[1], Sebastian Borchert[2], and Hauke Schmidt[3]

[1]Leibniz Institute of Atmospheric Physics at the University of Rostock, Kühlungsborn, Germany
[2]Deutscher Wetterdienst, Offenbach am Main, Germany
[3]Max Planck Institute for Meteorology, Hamburg, Germany
[*]now at Centre for Artificial Intelligence in Public Health Research (ZKI-PH), Robert Koch Institute, Berlin, Germany

**Correspondence:** Markus Kunze (kunze@iap-kborn.de)

**Abstract.** The Icosahedral Nonhydrostatic (ICON) general circulation model with upper atmosphere extension (UA-ICON) in the configuration with the physics package for numerical weather prediction (NWP) is presented with optimized parameter settings for the non-orographic and orographic gravity wave drag parameterizations (GWD). In this paper, we present UA-ICON(NWP) (version: ua-icon-2.1) in which we implemented optimized parameter settings for the GWD parameterizations to achieve more realistic MLT temperatures and zonal winds. The parameter optimization is based on perpetual January simulations targeting the thermal and dynamic state of the MLT and the Northern Hemisphere stratosphere. The climatology and variability of the Northern Hemisphere stratospheric winter circulation widely improve when applying UA-ICON with the NWP physics package compared to UA-ICON with ECHAM physics. Likewise improves the thermal and dynamic state of the MLT of the re-tuned UA-ICON(NWP) compared with the UA-ICON(NWP) using default settings. For UA-ICON(NWP), a statistical evaluation reveals a slight improvement in the stratosphere/mesosphere coupling compared to UA-ICON(ECHAM). The cold summer mesopause, the warm winter stratopause, and the related wind reversals are reasonably simulated. The GWD parameter optimization further significantly improves the frequency of major sudden stratospheric warmings (SSWs). However, the seasonal distribution needs improvement and the relative frequency of split vortex SSWs is underestimated compared to reanalyses, as is the zonal wavenumber 2 preconditioning of SSWs. This indicates that zonal wavenumber 2 forcing in UA-ICON(NWP) is underrepresented. The analysis of migrating diurnal and semidiurnal tides in temperature shows a good agreement of UA-ICON(NWP) with SABER-derived tides and the enhancement of the migrating semidiurnal tide during SSWs is well represented in UA-ICON(NWP).

## 1 Introduction

The mesopause region is the transition region between the middle and upper atmosphere. Its circulation is mainly driven by breaking gravity waves and the associated momentum flux divergence, driving the meridional residual circulation resulting in temperatures far away from radiative equilibrium in the mesosphere/lower thermosphere (MLT). This is reflected in the low atmospheric temperatures in the summer mesopause region with monthly mean minima below 140 K. These very low





temperatures enable phenomena such as the noctilucent clouds (NLCs) in high latitudes at an altitude between 82 and 85 km (e.g., Berger and von Zahn, 2002). General circulation models (GCMs) which include the MLT are, e.g., WACCM (Richter

et al., 2008; Gettelman et al., 2019), WACCM-X (Liu et al., 2018a), GAIA (Jin et al., 2012), the extended CMAM (eCMAM) (Beagley et al., 2000; Fomichev et al., 2002), HAMMONIA (Schmidt et al., 2006), HIAMCM (Becker and Vadas, 2020), KMCM (Becker, 2009), JAGUAR (Watanabe and Miyahara, 2009) and UA-ICON (Borchert et al., 2019), the upper atmosphere extension of the Icosahedral Nonhydrostatic (ICON) general circulation model (Zängl et al., 2015).

Concerning physics, ICON is equipped with two different packages. One is based on physics from the ECHAM model and

is now further developed at the Max Planck Institute for Meteorology (MPI-M). The other package, NWP, is maintained by the German Meteorological Service (Deutscher Wetterdienst, DWD). For applications with low horizontal resolutions relying on gravity wave parameterization, as presented here, the NWP package is the only choice in the current development of the ICON model. The initial UA-ICON version, as presented by Borchert et al. (2019), included both physics packages, however, the majority of the results are based on the ECHAM physics package which is no longer part of the actual development. Instead,

we use the NWP package here to validate its performance in the mesopause region, which has not been done so far.

An outstanding feature of ICON, and by this of UA-ICON, is its capability to apply several nests of successively higher horizontal resolutions embedded in a global coarser grid. Among the GCMs with extensions into the thermosphere, applying flexible subsequent grid refinements is a unique feature of UA-ICON. In the recent WACCM development with regional grid refinement (WACCM-RR), one finer grid is embedded in the global grid. Related studies at different model resolutions

using UA-ICON are available with the application of both physics packages. For UA-ICON(ECHAM) these include Stephan et al. (2020), analysing oblique gravity wave (GW) propagation during one minor SSW in ~20 km (referred to as R2B7 grid) simulations without GW-drag (GWD) parameterization. Applications of UA-ICON(ECHAM) in its original horizontal mesh size of ~160 km (R2B4), as evaluated by Borchert et al. (2019), are given in Stober et al. (2021) and Wallis et al. (2023). Stober et al. (2021) used several GCMs with upper atmosphere extensions (amongst these UA-ICON(ECHAM)) to

evaluate the performance of the models concerning winds and tides in the MLT, in comparison to winds derived from meteor radar at northern and southern latitudes, and to analyse inter-hemispheric coupling. In this comparison, the free-running UA-ICON performed relatively well in modelling the MLT wind fields in comparison to the otherwise nudged models. Recently, Wallis et al. (2023) used UA-ICON(ECHAM) to analyse the effect of an idealized large tropical volcanic eruption in June on the temperature structure in the mesosphere. Applications using the UA-ICON(NWP) configuration at a coarse horizontal

resolution of ~160 km (R2B4) are provided in Karami et al. (2022, 2023); Kim and Achatz (2021); Kim et al. (2021); Bölöni et al. (2021). They have either used the standard non-orographic gravity wave drag (NGWD) parameterization (Orr et al., 2010; Warner and McIntyre, 1996), namely Karami et al. (2022, 2023), or introduced a new NGWD parameterization overcoming the steady-state assumptions currently used in standard NGWD parameterizations (Bölöni et al., 2021; Kim et al., 2021; Kim and Achatz, 2021). In our tuning experiments, we also consider the effects of orographic gravity waves (OGW) which directly

influence the stratospheric vortex and by this, indirectly, the mesospheric circulation. Higher horizontal resolutions of UA-ICON(NWP) with a global ~20 km grid spacing (R2B7), and the application of two additional nests with higher horizontal





resolutions of ∼10 km (R2B8) and ∼5 km (R2B9), respectively, were used in the study of Charuvil Asokan et al. (2022) for the validation of vertical and horizontal winds derived from meteor radar observations.

The main objective of this work is to document the tuning requirements of the GWD parameterization for UA-ICON(NWP),
necessary for its application with the coarse horizontal resolution of R2B4. Simulations with a coarse resolution are a prerequisite for short-term experiments with higher-resolution nests and long-term climatological studies. We use several climatologies, mainly based on satellites, as references for these global simulations. This is done for the primary meteorological quantities like wind and temperature. More dedicated quantities like wave drag are compared with other models. Besides the seasonal cycle of the middle atmosphere, we also investigate major sudden stratospheric warmings during the NH winter season (in the
following abbreviated with SSWs), which are the most dramatic changes in the stratosphere with large temperature changes in the polar region within a couple of days (Scherhag, 1952). A large increase in temperature in polar regions also affects the zonal wind which is decelerated, and in the case of major SSWs can turn from the usual zonal mean eastward-directed flow to a zonal mean westward-directed flow. SSWs are forced by the dissipation of upward propagating large-scale planetary (Rossby) waves (Charney and Drazin, 1961). The reviews of Baldwin et al. (2021) and Butchart (2022) summarize the research and progress
in understanding the causes and consequences of SSWs over the past 70 years. A satellite-based benchmark test, including the mesosphere, was developed by Zülicke et al. (2018) and will be used in the present study to analyse SSW effects.

Finally, we identify atmospheric tides in UA-ICON and their behaviour concerning SSWs. Atmospheric tides are global-scale oscillations with a period of a day and its harmonics (Chapman and Lindzen, 1969). Solar tides are thermally driven by periodic heating due to the absorption of solar radiation mainly by water vapor in the troposphere and ozone in the stratosphere (Forbes
and Garrett, 1979), as well as due to latent heat release in the deep tropical convection (Zhang et al., 2010a, b). Tides generated in the troposphere and stratosphere grow in amplitude as they propagate vertically into the MLT region where they achieve their maximum amplitudes (Hagan and Forbes, 2002, 2003; Oberheide et al., 2011). MLT tides are important in driving the upper atmosphere including the ionosphere (Yamazaki and Richmond, 2013; Jones et al., 2014). Global observations from satellites have established that westward-propagating solar 'migrating' (i.e., Sun-synchronous) diurnal (24 h) and semidiurnal
(12 h) tides are the dominant modes of MLT tides (McLandress et al., 1996; Forbes et al., 2008; Yamazaki et al., 2023). Thus, we will focus on these tidal modes.

We introduce the UA-ICON model and the data used for evaluation in Sect. 2. In Sect. 3 we evaluate the seasonal averages of the zonal wind and temperature around the solstices for UA-ICON(NWP) after optimizing the GWD parameterizations. Section 4 summarizes the tuning process for the GWD parameterizations performed in a perpetual January setup. We document
the detailed impact of both OGWD and NGWD. An evaluation of the northern hemispheric winter variability and the statistics of major SSWs are presented in Sect. 5. Sect. 6 is devoted to the simulation of thermal tides compared to SABER observations. The final Sect. 7 concludes with a discussion and a summary.


## 2   Model, data and methods

### 2.1   Model and setup

ICON is a joint development of the DWD, the MPI-M, Deutsches Klimarechenzentrum (DKRZ), the Karlsruhe Institute of
Technology (KIT), and the Center for Climate Systems Modeling (C2SM) to create a modelling framework that serves the
need for numerical weather prediction (NWP) as well as the application for climate simulations. For these tasks originally two
physics packages with sets of parameterizations have been available, namely NWP (Zängl et al., 2015) for weather prediction
and ECHAM for climate applications (Giorgetta et al., 2018; Jungclaus et al., 2022). ICON employs a non-hydrostatic dynam-
ical core on an unstructured triangular C-grid, and a geometric altitude grid, which is terrain-following up to a certain height.
A Lorenz-type staggering is applied in the vertical with the prognostic variables of the grid-edge-normal wind components,

| Process | ua-icon-1.02 (ECHAM) | ua-icon-2.1 (NWP) |
|---|---|---|
| Longwave (LW) and Shortwave (SW) radiation | PSRAD (Pincus and Stevens, 2013), (based on RRTM, Mlawer et al., 1997) | ecRAD (Hogan and Bozzo, 2016), (based on RRTM, Mlawer et al., 1997) |
| Turbulent transfer | ECHAM6.3 Pithan et al. (2015) | Prognostic turbulence kinetic energy (TKE) (COSMO, default) Raschendorfer (2001) |
| Cloud cover | Diagnostic Sundqvist et al. (1989) | Diagnostic probability density function (PDF) M. Köhler et al. (DWD) (default) |
| Convection | Tiedtke (1989)/Nordeng (1994) | Tiedtke (1989)/Bechtold et al. (2008) (default) |
| Cloud microphysics | Single-moment scheme Lohmann and Roeckner (1996) | Single-moment scheme (default) Doms et al. (2011), Seifert (2008) |
| Non-orographic GWD | Hines (1997a, b) | Orr et al. (2010) (IFS), (based on Warner and McIntyre, 1996) |
| Subgrid scale orographic effects (SSO) | Lott and Miller (1997); Lott (1999) | Lott and Miller (1997) (COSMO, default) |

**Table 1.** Parameterizations for physical processes given in the first column in UA-ICON simulations with ECHAM physics (second column) and NWP physics package (third column).

the potential temperature, and the density of moist air defined on full model levels and the vertical wind component defined on
the half levels. The main parameterizations of the NWP and ECHAM physics packages, as applied in this work, are listed in
Table 1. ICON's upper atmosphere (UA) extension consists of an optional deep-atmosphere dynamical core and supplementary
physical parameterizations for the relevant physical processes in the MLT (Borchert et al., 2019). These are parameterizations
for molecular diffusion (Huang et al., 1998), ion-drag and Joule heating (Hong and Lindzen, 1976), frictional heating (Gill,
1982), the heating rates of $O_2$ absorption of ultraviolet (UV) in the Schumann-Runge bands and continuum (Strobel, 1978),
absorption of extreme-UV by $N_2$, O, and $O_2$ (Richards et al., 1994), non-LTE infrared cooling by $CO_2$, NO, and $O_3$ (Fomichev
and Blanchet, 1995; Fomichev et al., 1998; Ogibalov and Fomichev, 2003), and infrared NO cooling at $5.3\mu$m (Kockarts,



1980). UA-ICON does not include interactive chemistry and therefore describes the chemical heating with a climatological annual cycle from a 30-year time slice simulation of HAMMONIA (Schmidt et al., 2006).

The ICON release icon-2024.01-1 (ICON partnership (DWD, MPI-M, DKRZ, KIT, and C2SM), 2024) is the code basis in this work whenever using the NWP physics package (ua-icon-2.1), whereas the UA-ICON simulation using the ECHAM physics package is based on a slightly updated version of Borchert et al. (2019) (ua-icon-1.02).

All simulations in this work use the R2B4 horizontal resolution, corresponding to a grid distance of ∼160 km, and 120 levels up to 150 km. They all apply the upper-atmosphere (UA) extension of ICON, described by Borchert et al. (2019) for the ECHAM and the NWP physics packages. Three time-slice simulations with a seasonal cycle are presented here with boundary

| Simulation | Version (Git tag) | Physics | Years | Setup |
|---|---|---|---|---|
| UA-ICON(ECHAM) | ua-icon-1.02 | ECHAM | 20 | Borchert et al. (2019) |
| UA-ICON(NWPD) | ua-icon-2.1 | NWP | 60 | F1C1 |
| UA-ICON(NWP) | ua-icon-2.1 | NWP | 60 | F2C30-S |

**Table 2.** UA-ICON time slice simulations with a seasonal cycle, with the columns indicating the simulation's label, the version, the physics package, the number of years after a one-year spin-up, and the setup. Table 3 details the simulation's setup with NWP physics.

conditions representative of the late 1990s, as listed in Table 2. The UA-ICON(NWP) simulations use repeated, seasonally varying climatological (1979–2016) conditions for the sea-surface temperatures and sea-ice concentrations from the Program

for Climate Model Diagnosis and Intercomparison (PCMDI) Atmospheric Model Intercomparison Project (PCMDI-AMIP 1.1.2) dataset (Taylor et al., 2000). The radiatively active gases in ecRAD are prescribed as globally yearly averaged values (1990-2000) for $CO_2$, and modified with a $\tanh$ profile for $CH_4$, $N_2O$, CFC-11, and CFC-12. Tropospheric background aerosol optical properties, representing conditions of the year 1865, are prescribed for ecRAD (Kinne et al., 2013). The radiatively active gases for the upper-atmosphere extension, namely $CO_2$, NO, $O_3$, $O_2$, and O, and the $O_3$ for ecRad are prescribed from

a 35-year climatology of a HAMMONIA simulation (Schmidt et al., 2006). The solar forcing is constant with 14 spectrally resolved irradiances and a total solar irradiance of 1361.12 W m$^{-2}$, averaged from 1979–2016, using the dataset prepared for CMIP6 (Matthes et al., 2017); the F10.7 cm solar flux for the calculation of the EUV heating rates is set to 150 sfu (1 sfu = $1 \times 10^{-22}$ W m$^{-2}$ Hz$^{-1}$). The 20-year UA-ICON(ECHAM) simulation uses the same boundary conditions and settings as Borchert et al. (2019), the UA-ICON(NWPD), and UA-ICON(NWP) simulations all use NWP physics. They are run for

60 years after one year of spin-up. UA-ICON with NWP physics requires longer simulation periods, than UA-ICON with ECHAM physics, e.g. for reliable conclusions concerning statistics based on major SSWs, as the dynamic variability is much larger. The second simulation, UA-ICON(NWPD), uses the default settings (therefore labelled NWPD) for the OGWD and NGWD parameterizations (Label F1C1 in Table 3) and UA-ICON(NWP) uses tuned parameters for the OGWD and NGWD parameterizations (Label F2C30-S in Table 3). The major features of ECHAM and NWP physics are summarized in Table 1.

Especially the difference in NGWD parameterization between the physics packages, Hines (1997a, b) in ECHAM and Warner and McIntyre (1996) in NWP, has a substantial impact on the climatology of the MLT, which is emphasized in Sect. 3.





## 2.2 Reanalyses and satellite observations

We use temperature and zonal wind of the three reanalyses ERA-5 (Hersbach et al., 2020), NCEP/NCAR (Kalnay et al., 1996; Kistler et al., 2001), and MERRA2 (Gelaro et al., 2017) for the evaluation of the NH stratospheric variability and the statistical
evaluation of SSWs.

For the evaluation of the MLT temperature and zonal wind, we use the latest version (v2.07, v2.08 from December 2020 onward) of satellite observations from the SABER (Sounding of the Atmosphere using Broadband Emission Radiometry) instrument on the TIMED (Thermosphere, Ionosphere, Mesosphere, Energetics and Dynamics) satellite (Russell III et al., 1999; Dawkins et al., 2018). The original level 2A data, sorted by event and altitude, is binned to a regular latitude-by-altitude
grid with a temporal resolution of one month and a spatial resolution of 5° horizontally and 1 km vertically. As a reference for the zonal mean zonal wind in the MLT, we use data from the UARS (Upper Atmosphere Research Satellite) Reference Atmosphere Project (URAP) (Swinbank and Ortland, 2003). For daily resolved temperature observations through the middle atmosphere, including the MLT region, we use Aura Microwave Limb Sounder (Aura-MLS) version 5.0 Level 3 data on pressure levels (Livesey et al., 2022).

## 2.3 Methods

The original UA-ICON output data are stored on the R2B4 triangular grid with an output frequency of six hours, instantaneously for the basic dynamical quantities. Additionally, temperature, pressure, and zonal and meridional wind components are output instantaneously at a one-hour frequency for the analyses of tidal activity. The model daily averaged tendencies of the physical parameterizations are output with a frequency of one day. These triangular output data are transferred to a regular
Gaussian T63 grid with 192 longitudes and 96 latitudes, and 120 fixed geometric altitude levels, corresponding to the model full height levels once the influence of the orography levels off, for the post-processing procedures. The so-called transformed Eulerian mean (TEM) quantities (Andrews and McIntyre, 1976; Andrews et al., 1987), namely the Eliassen–Palm (EP) diagnostics, which are the EP-Flux ($\mathbf{F}$) and its divergence ($\nabla \cdot \mathbf{F}$), and the meridional and vertical components of the residual mean meridional circulation (MMC) ($\overline{v}^*$, $\overline{w}^*$) are calculated on the 120 height levels of the T63-grid. We use the formulation of the
hydrostatic primitive equations (HPE) on geometric coordinates (HPE(z)) of Hardiman et al. (2010) for the computation of the EP diagnostics. Hardiman et al. (2010) demonstrated the large error made when calculating the EP diagnostics based on the formulation of the HPE on log-pressure coordinates HPE(ln(p)) for non-hydrostatic models formulated on geometric altitude levels. For the analysis of major sudden stratospheric warmings and the related diagnostics of mesospheric coupling, the daily UA-ICON output is vertically interpolated to a set of 53 standard pressure levels. This allows a more direct comparison with
the reanalysis products and other published related benchmarks.



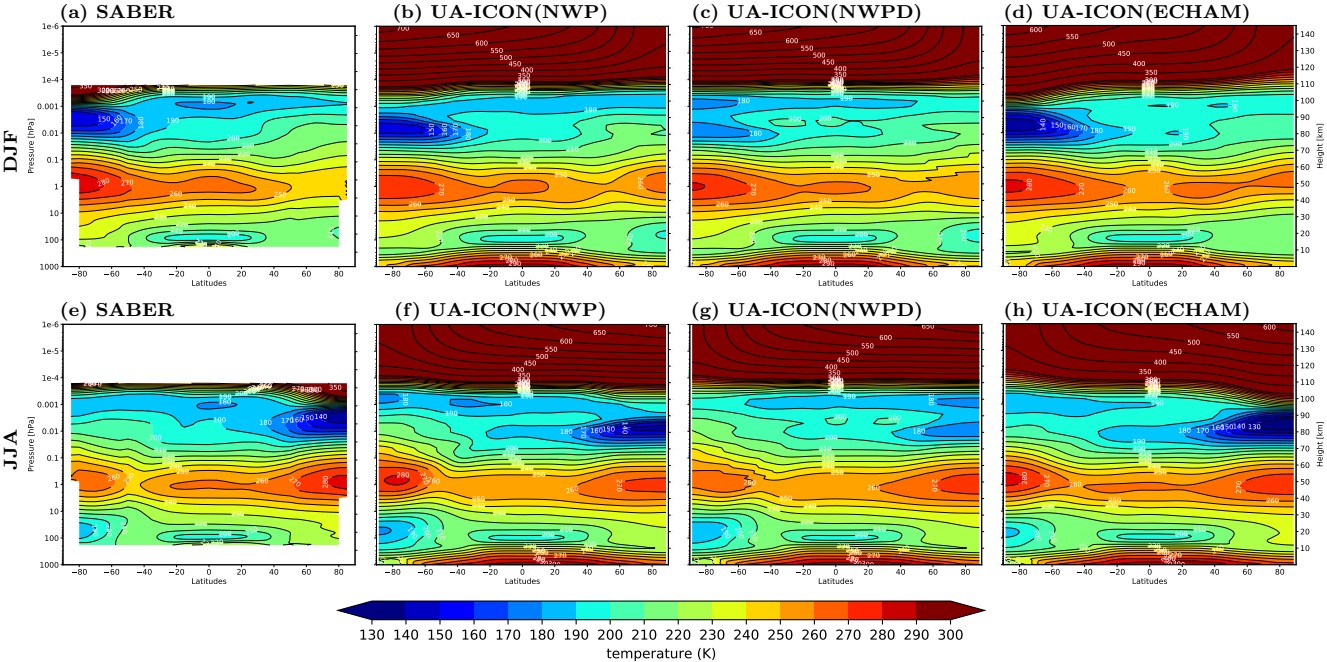

**Figure 1.** Climatology of zonal mean temperature for Dec./Jan./Feb. (top row a–d) and Jun./Jul./Aug. (bottom row e–h) seasonal mean; (a, e) SABER (2002–2022), (b, f) UA-ICON with NWP physics and tuned gravity waves (F2C30-S, in table 3); (c, g) UA-ICON with NWP physics (default settings, F1C1 in table 3); (d, h) UA-ICON with ECHAM physics.

## 3 Global circulation

We compare the zonal mean temperature (Fig. 1) and zonal mean zonal wind (Fig. 2) in the boreal and austral winter seasons from UA-ICON to SABER and URAP climatologies. Comparing UA-ICON(NWP) with its default parameter settings for the GW parameterizations, chosen for the standard version up to ∼75 km (Fig. 1, 2, c, g, in the following abbreviated with UA-ICON(NWPD)) with temperature from SABER (Fig. 1,a,e) and the URAP zonal wind (Fig. 2,a,e), we can identify the deficits in the MLT region with a warm temperature bias in the summer mesopause region of more than 30 K and an eastward zonal mean zonal wind extending from the middle atmosphere to the lower thermosphere during the winter seasons and a westward zonal mean zonal wind during the summer seasons reversing only slightly to an eastward direction in an altitude from 100 to 110 km. We significantly reduce these deficits of UA-ICON(NWPD) in the MLT region concerning the temperature, and the zonal wind with the tuned parameters for the OGWD and NGWD parameterizations of UA-ICON(NWP) (Fig. 1, 2, b, f). Details of the tuning procedure and the chosen parameters follow in Sect. 4. The temperature in the austral and boreal summer MLT region is decreasing by more than 30 K in UA-ICON(NWP), now comparable to SABER (Fig. 1, a and e) during austral and boreal summer seasons, whereas UA-ICON(ECHAM) is showing a cold bias of 10 K in this region for both seasons (Fig. 1, d, h). The reversals from the westward-directed summer zonal mean zonal wind in the middle atmosphere to an eastward zonal wind direction in the upper mesosphere are present in UA-ICON(NWP) (Fig. 2, b and f) with a magnitude stronger compared



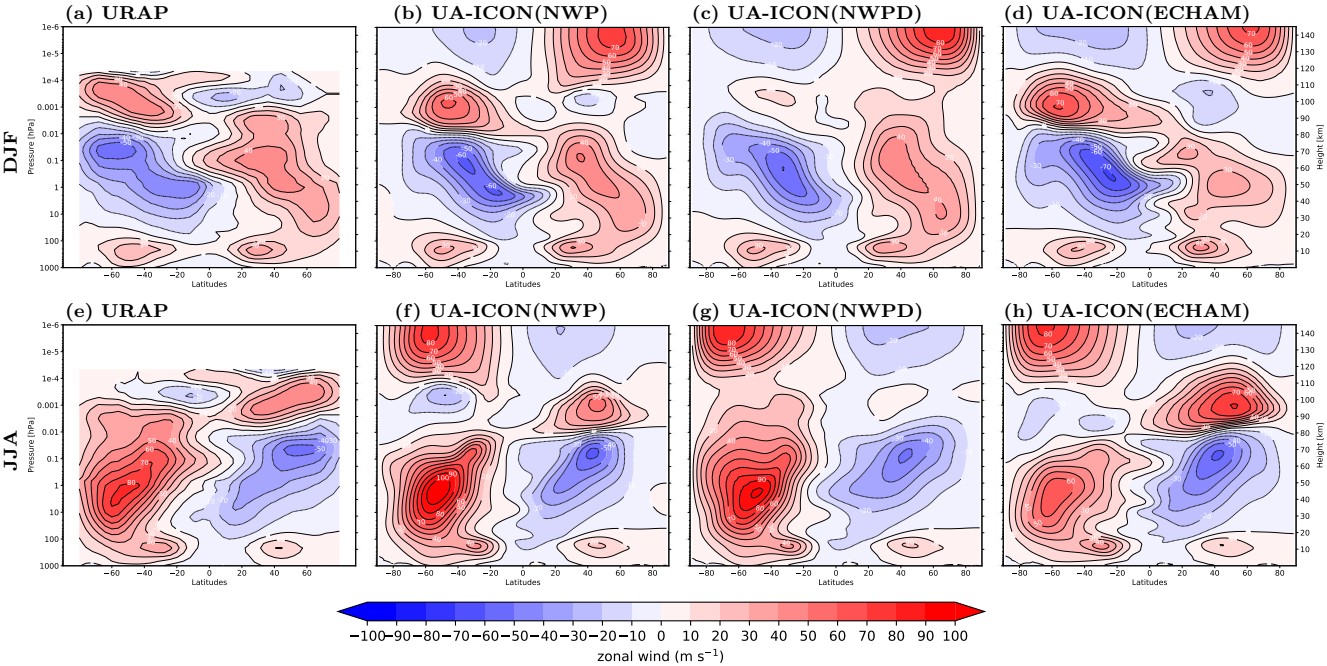

**Figure 2.** Climatology of zonal mean zonal wind for Dec./Jan./Feb. (top row a–d) and Jun./Jul./Aug. (bottom row e–h) seasonal mean; (a, e) URAP climatology; (b, f) UA-ICON with NWP physics and tuned gravity waves (F2C30-S, in table 3); (c, g) UA-ICON with NWP physics (default settings, F1C1 in table 3); (d, h) UA-ICON with ECHAM physics.

to URAP (Fig. 2, a and e), whereas UA-ICON(ECHAM) (Fig. 2, d and h) shows a wind reversal too intense and too low in altitude as reported by Borchert et al. (2019). This effect of the NGWD tuning in UA-ICON(NWP) limits the mesospheric polar vortex to an altitude of ~80 km, whereas in both winter seasons, it extends to an altitude of approximately 100 km in the URAP climatology. Comparable differences were reported, e.g., by Harvey et al. (2019) in modelling the extension of the mesospheric

polar vortex with WACCM, although specifying the dynamics up to 60 km from MERRA2 data. The winter westerly winds at high latitudes in WACCM extend to lower altitudes, compared to geostrophic zonal winds calculated from SABER-derived geopotential heights. The magnitude of the eastward-directed zonal winter circulation in the middle atmosphere is too high in the SH in UA-ICON(NWP) by 20 m s$^{-1}$, and slightly too high in the NH. However, the position of the westerly jets is well captured in both hemispheres compared to URAP. A second side effect of increasing the eastward-directed NGWD in

UA-ICON(NWP) is the weakening of the westward-directed zonal circulation in summer, leading to a shift of the -30 m s$^{-1}$ contour to lower latitudes by more than $10°$ in the upper mesosphere. The magnitude of the NH summer westward-directed zonal circulation (<-50 m s$^{-1}$) compares well with URAP, however, the location of this easterly jet is too low in altitude and shifted too far to low latitudes. This shift of the easterly jet appears as well in the SH summer together with a slightly too strong westward-directed zonal circulation (<-60 m s$^{-1}$).





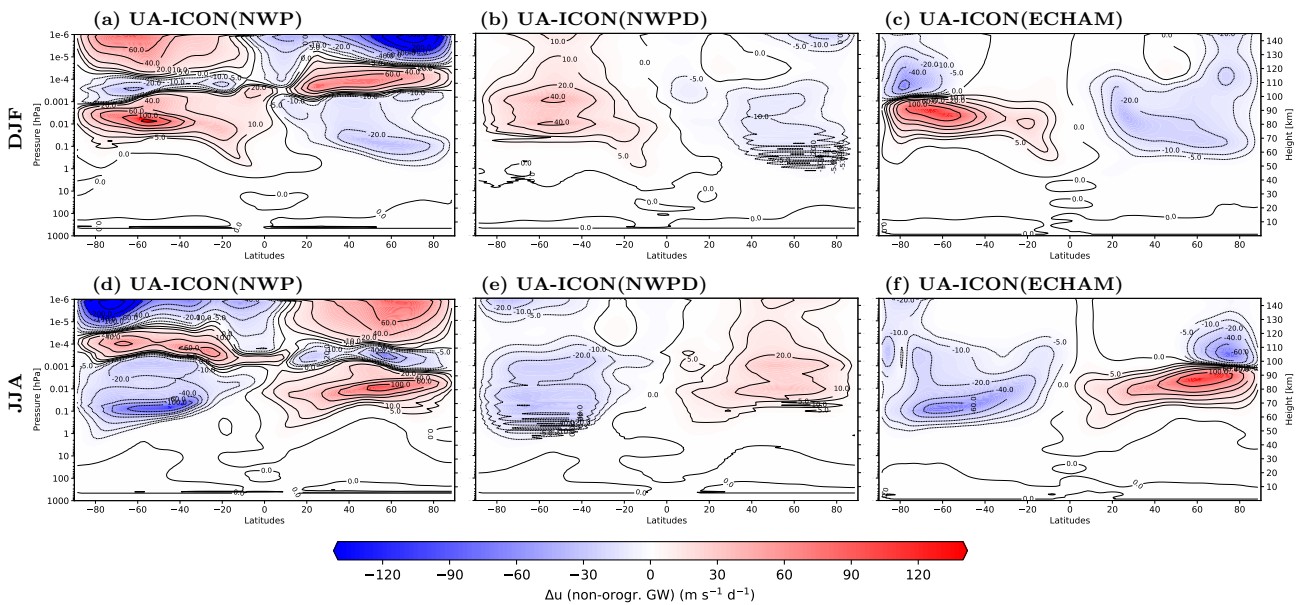

**Figure 3.** Multi-year boreal (top row a–c) and austral (bottom row d–f) winter seasonal, zonal mean zonal wind tendencies in m s$^{-1}$ d$^{-1}$ due to non-orographic gravity waves of UA-ICON simulations with NWP physics (Warner and McIntyre, 1996) with tuned gravity waves (a, d), NWP physics with default gravity wave parameters (b, e), and ECHAM physics (Hines, 1997a, b) (c, f).

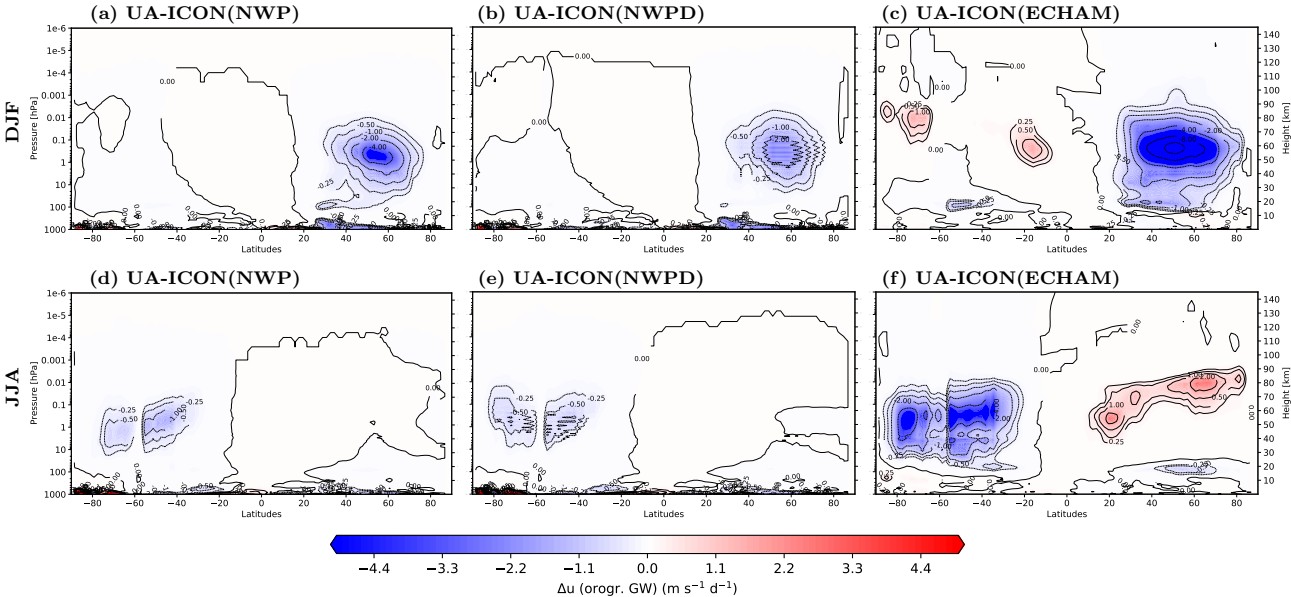

**Figure 4.** Multi-year boreal (top row a–c) and austral (bottom row d–f) winter seasonal, zonal mean zonal wind tendencies in m s$^{-1}$ d$^{-1}$ due to orographic gravity waves of UA-ICON simulations with NWP physics (Lott and Miller, 1997) with tuned non-orographic gravity waves (a, d), NWP physics with default gravity wave parameters (b, e), and ECHAM physics (Lott, 1999) (c, f).





UA-ICON(ECHAM) and UA-ICON(NWPD) have a warm bias in winter stratopause temperature compared to SABER, which intensifies by the tuning of the NGWD parameterization in UA-ICON(NWP). However, the winter stratospheric temperature is in better agreement with SABER for both UA-ICON(NWP) and UA-ICON(NWPD), and the strength of the northern hemispheric stratospheric vortex is much better captured compared with URAP. In general, using ICON with the NWP physics package leads to a much better representation of the stratosphere that eliminates the problems with the stratospheric circulation

and temperatures in ICON(ECHAM) noticed by Giorgetta et al. (2018) and for UA-ICON(ECHAM) by Borchert et al. (2019).

    The main driver of the mesosphere global circulation is the breaking of NGWs (e.g., Becker, 2012; Vincent, 2015, and references therein) that force a meridional circulation from the summer hemisphere to the winter hemisphere, downwelling and adiabatic warming over the winter pole, and in turn upwelling in the summer mesosphere leading to adiabatic cooling with temperatures far below the radiative equilibrium. The coarse resolution of the UA-ICON simulations in the present work

does not allow for explicitly resolving the upward propagation and breaking of NGWs. Therefore, the effects of NGWs are parameterized according to Hines (1997a, b) (H97) in UA-ICON(ECHAM) and based on Warner and McIntyre (1996) (WM96) in UA-ICON(NWP). Figure 3 shows the zonal mean zonal wind tendencies due to the parameterized NGWD. Compared to the H97 tuning, with more than 130 m s$^{-1}$ d$^{-1}$ in both winter seasons in an altitude between 80 and 100 km (Fig. 3, c, f), WM96, with NWP default settings, shows much smaller tendencies only up to 40 m s$^{-1}$ d$^{-1}$ in SH winter or 24 m s$^{-1}$ d$^{-1}$ in NH winter

(Fig. 3, b, e). This missing NGWD in the MLT is mainly responsible for the large discrepancies in climatological zonal mean zonal wind and temperatures as present in UA-ICON(NWPD) (Fig. 2, 1, c, g) and by increasing the NGWD of WM96 in the MLT to values comparable to H97 (Fig. 3, a, d) the temperature and zonal wind in the MLT improve considerably (Fig. 1, 2, b, f). The latitudinal structure of the zonal wind tendencies from the H97 and WM96 up to an altitude of 100 km are very similar. This similarity is the result of increasing the saturation momentum flux density of WM96, as proposed by McLandress and

Scinocca (2005) by a factor of 30 (see Sect. 4 for more details). Both physics packages in ICON use the parameterization based on Lott and Miller (1997) (LM97) to account for the orographic gravity wave drag (OGWD). However, the ECHAM physics package uses the implementation of ECHAM6 (Lott, 1999), whereas NWP physics uses the COSMO implementation. OGWD is acting mainly in the stratosphere, and the resulting zonal mean zonal wind tendencies show relatively huge differences of up to a factor of four, with larger OGWD in UA-ICON(ECHAM) (Fig. 4, c, f) than in UA-ICON(NWPD) (Fig. 4, b, e) and UA-

ICON(NWP) (Fig. 4, a, d). The parameters of the OGWD parameterization in UA-ICON(NWP) are adapted for the coarser R2B4 horizontal resolution of UA-ICON as listed in Table 3 (F2C30-S). By this, the OGWD increases in the NH middle atmosphere in winter by a factor of two (UA-ICON(NWP), Fig. 4, a), whereas there are only minor changes in JJA (Fig. 4, d).

## 4  Gravity wave tuning

The presented shortcomings in the UA-ICON(NWPD) (with default settings) climatology of the MLT are the main motivation

for a re-tuning of the WM96 NGWD and LM97 OGWD parameterizations in UA-ICON(NWP). We perform a series of perpetual January simulations (Table 3) with variations of the scaling factor of the saturation momentum flux density ($C^*$) of WM96, introduced by McLandress and Scinocca (2005) when comparing H97, WM96, and Alexander and Dunkerton (1999)

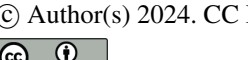



**Figure 5.** Vertical profiles of quantities averaged for specific regions in January for (left column) the Southern hemisphere; (right column) the Northern hemisphere; from top to bottom for the temperature in K; the zonal wind in m s$^{-1}$; the transformed Eulerian mean meridional velocity ($\overline{v}^*$) in m s$^{-1}$; the zonal wind tendency due to non-orographic gravity waves in m s$^{-1}$ d$^{-1}$; and the divergence of the Eliassen-Palm vector ($\nabla \cdot \mathbf{F}$) in m s$^{-1}$ d$^{-1}$.



| Label | $C^*$ | $\rho_0\|\hat{\mathbf{F}}_p\|$ | $K_{wake}$ | $K_{drag}$ | $Fr_{crit}$ |
|---|---|---|---|---|---|
| NoOGWD | 1.0 | 2.50 | - | - | - |
| NoNGWD | - | - | 1.5 | 0.075 | 0.4 |
| NoGWD | - | - | - | - | - |
| **F1C1 (NWPD)** | **1.0** | **2.50** | **1.5** | **0.075** | **0.4** |
| F2C1 | 1.0 | 1.75 | 1.5 | 0.075 | 0.4 |
| F2C10 | 10.0 | 1.75 | 1.5 | 0.075 | 0.4 |
| F2C20 | 20.0 | 1.75 | 1.5 | 0.075 | 0.4 |
| F2C30 | 30.0 | 1.75 | 1.5 | 0.075 | 0.4 |
| **F2C30-S (NWP)** | **30.0** | **1.75** | **1.1** | **0.052** | **0.6** |

**Table 3.** Parameter setting of the WM96 NGWD ($C^*$, $\rho_0\|\hat{\mathbf{F}}_p\|$) and the LM97 OGWD parameterizations of the tuning simulations in perpetual January mode; $C^*$, a factor to increase the saturation momentum flux density; $\rho_0\|\hat{\mathbf{F}}_p\|$, the total launch momentum flux in each azimuth in mPa; $K_{wake}$, the low-level wake drag constant; $K_{drag}$, the gravity wave drag constant; $Fr_{crit}$, the critical Froude number. All simulations use the same default settings for the WM96 tunable $L_p$ = 450 hPa, the launch height of the gravity wave spectrum. The setups in bold are used for the UA-ICON(NWPD) and UA-ICON(NWP) simulations with a seasonal cycle.

NGWD parameterizations. The total launch momentum flux in each azimuth ($\rho|\hat{F}_p|$) is tested with its default value of 2.50 mPa and compared with 1.75 mPa (setups labelled with F1 and F2, respectively), and the launch height ($L_p$ = 450 hPa) stays with

its default value. The simulation labelled with "-S" includes changes in tunable parameters of the OGWD parameterization, namely the low-level wake drag constant ($K_{wake}$), the gravity wave drag constant ($K_{drag}$), and the critical Froude number ($Fr_{crit}$).

Figure 5 shows profiles of averaged quantities for the Northern (NH) and Southern (SH) hemispheres, to evaluate the effect of parameter changes. As shown in Sect. 3, the default WM96 settings lead to insufficient NGWD in the MLT region. By

switching off the WM96 NGWD completely (NoNGWD), we can evaluate the effect of the parameterized NGWs, also shown with the latitude-altitude sections (Fig. 6) of the anomalies NoNGWD minus F1C1 (left column). NoNGWD shows a strong response in the profiles for all averaged quantities, with the summer mesopause being too high in altitude and temperatures lower than the default (F1C1) case, but still too warm compared to SABER. The SH easterly wind regime extends to the lower thermosphere, and the meridional component of the residual MMC ($\overline{v}^*$) is even stronger northward directed than for

F1C1, peaking near 100 km globally. The EP-flux divergence ($\nabla \cdot \mathbf{F}$) reflects the forcing of the zonal mean zonal wind by resolved waves. Without NGWD (NoNGWD) $\nabla \cdot \mathbf{F}$ is more intense and in the opposite direction than F1C1, indicating an increase in dissipating resolved waves. These are probably GW with an eastward-directed phase speed, which can propagate to considerably higher altitudes in the easterly wind regime extending to the MLT region in the SH. For the simulation F2C1, the momentum flux $\rho|\hat{F}_p|$, launched near 450 hPa is decreased by 30%. With this, the zonal wind tendency due to NGWD

decreases, and all other quantities slightly change in a direction discussed for NoNGWD, tending to worsen the climatology in





the MLT. However, for simulations with an increasing C* a smaller $\rho|\hat{F}_p|$ shows better results in the MLT. In the simulations F2C10–F2C30, the parameterized WM96 NGWD increases by increasing the saturated momentum flux density from its default by a factor varying from C* = 10 to C* = 30, which effectively increases the altitude where the upward propagating NGWs dissipate with a direct impact on the tendency of the zonal wind in the MLT region calculated by the NGWD parameterization

(ΔU-NGWD). The weak eastward-directed ΔU-NGWD of 40 m s$^{-1}$ d$^{-1}$ in the SH of the default (F1C1, Fig. 5, and NWPD, Fig. 3(b)) changes to 80 m s$^{-1}$ d$^{-1}$ (F2C10) or to more than 110 m s$^{-1}$ d$^{-1}$ (F2C20, F2C30, Fig. 5, and NWP, Fig. 6) within a shallow layer, peaking near 81 km. The magnitude of ΔU-NGWD in F2C30 is comparable to ΔU-NGWD of H97, which peaks slightly higher at 84 km and extends to a broader region in altitude. Near 100 km, ΔU-NGWD changes to a westward direction in F2C20, F2C30, and in H97, with F2C20 and F2C30 showing a more narrow peak in ΔU-NGWD than H97.


The summer mesopause temperature and the zonal wind regimes in the MLT region are changing, forced by the strengthened NGWD (Figs. 5 and 6(right)). With increasing C*, the SH MLT temperature decreases in the simulations F2C10–F2C30, where the double peak structure of the mesopause in F1C1 and F2C1 is vanishing and only the lower part of the mesopause is persisting. The temperature is comparable to SABER but temperatures lower than 150 K concentrate in a narrow altitude

region at ∼80–85 km, lower than SABER showing these temperatures between ∼84–93 km. The zonal wind direction reverses from the westward-directed summer circulation in the SH stratosphere/mesosphere to eastward zonal wind and from the eastward-directed winter circulation in the NH to westward zonal wind in the MLT. The NH polar cap mesopause temperature is increasing, a negative side effect, as the default settings (F1C1) already show a mesopause temperature slightly warmer than SABER. The temperature changes are directly related to changes in the MMC. As a measure of the strength of the MMC, we

use $\overline{v}^*$ (Figs. 5 and 6), showing an increasing northward summer-to-winter directed flow with increasing C* in a layer around the mesopause. Above 100 km the MMC turns to a southward-directed flow with increasing C*, which is directly related to the more westward-directed NGWD-induced zonal wind changes in the SH and the change to an eastward-directed NGWD in the NH. This winter-to-summer directed flow extends over both hemispheres in a layer between 100 and 120 km with a minimum near the Equator, and is stronger in the NH for UA-ICON(NWP) than UA-ICON(ECHAM). Qian et al. (2017) and Wang et al.

(2022) reported similar features for SD-WACCM simulations and compared them to the MMC derived from vertical gradients of SABER $CO_2$ volume mixing ratio. They found large vertical $CO_2$ gradients in the 95–110 km height region at summer hemispheric polar latitudes consistent with the SD-WACCM $CO_2$ gradients, and the upward-directed residual flow in the upper mesosphere and the downward-directed residual flow of the lower thermosphere.

Besides the average temperature, the variability and the range of temperatures in the summer mesopause region and the NH

stratosphere are important measures. We use the probability density function (PDF) of daily January zonal mean temperature in selected areas near the NH stratopause, the NH lower stratosphere (Fig. 7), the SH mesopause, and stratopause (Fig. 8) to show the effect of GW parameter tuning of UA-ICON(NWP) in comparison to UA-ICON(ECHAM), SABER data, and ERA-5 reanalysis data. The temperature distribution near the stratopause (Fig. 7, a) confirms that all UA-ICON simulations are too warm, compared to SABER and ERA-5 when the non-orographic GW parametrization is on. The UA-ICON(NWP) simulations

without NGWD parameterization (NoNGWD, NoGWD) show a slightly lower average temperature and a wider distribution

**Figure 6.** Perpetual January zonal mean long-term mean anomalies calculated with respect to F1C1 (NWPD) for noNGWD (left row) and F2C30-S (NWP) (right row); (a) temperature in K; (b) zonal wind in m s$^{-1}$; (c) transformed Eulerian mean meridional velocity ($\overline{v}^*$) in m s$^{-1}$; (d) zonal wind tendency due to non-orographic gravity waves in m s$^{-1}$ d$^{-1}$; (e) divergence of the Eliassen-Palm vector ($\nabla \cdot \mathbf{F}$) in m s$^{-1}$ d$^{-1}$.



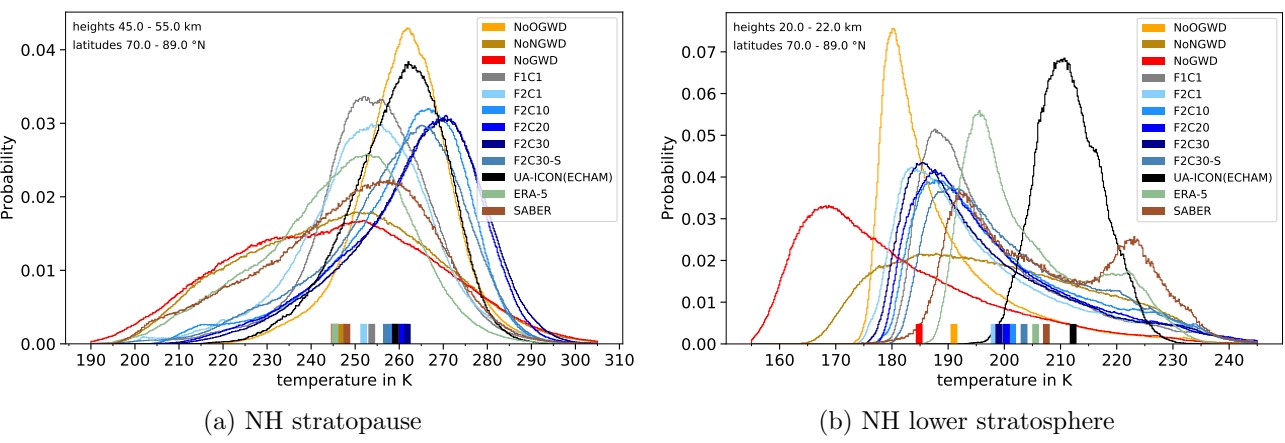

**Figure 7.** Probability density functions (PDF) of daily mean temperature, for ERA-5 and UA-ICON, and the original temperature soundings for SABER, in the northern hemisphere; (a) for the northern polar cap (70–89°N) stratopause region (45–55 km); (b) for the northern polar cap lower stratosphere (20–22 km).

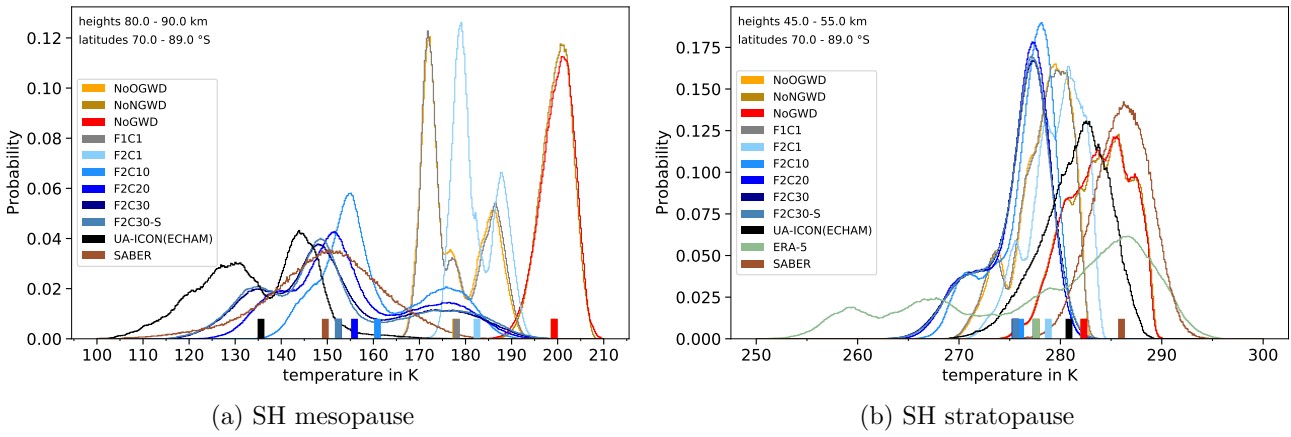

**Figure 8.** As Figure 7; (a) for the southern polar cap (70–89°S) mesopause region (80–90 km); (b) for the southern polar cap stratopause (45–55 km).





than ERA-5. With the increasing strength of the NGWD, the NH stratopause temperatures increase, where all UA-ICON simulations with NGWD parameterization show a more narrow probability density function. In the NH lower stratosphere (Fig. 7, b) the average temperature is much too low without OGWD parameterization (NoOGWD) and even lower without any GW parameterizations (NoGWD). Most UA-ICON(NWP) simulations, with default settings (F1C1) and an increase (F2C10–

F2C30) of the NGWD, show an average lower stratospheric temperature around 200 K, 4–8 K lower than ERA-5 and even more compared to the ∼2 K higher temperature from SABER. The simulation F2C30-S, including adapted parameters for the OGWD parametrization, shows the best agreement with ERA-5 temperature which is only slightly colder, whereas UA-ICON(ECHAM) shows a warm bias and a very narrow PDF, indicating a much too low inter-annual variability.

The temperatures in the SH mesopause region are the lowest in the Earth's atmosphere. The average SABER temperature is near 150 K with a range of temperatures from 110 to 180 K. Whereas UA-ICON(ECHAM) shows a cold bias in this region, the UA-ICON(NWP) simulations with default settings for the WM96 NGWD show a strong warm bias and the double peak structure of the mesopause temperature (Fig. 5) with a minimum near 90 km which also is reflected in the PDF, showing two distinct peaks. Decreasing $\rho|\hat{F}_p|$ by 30% (F2C1), slightly increases the temperatures, shifting the distribution towards the ones of NoNGWD and NoGWD, that, however, do not cover the region of the mesopause (∼100–110 km) in these simulations and therefore show an average temperature slightly below 200 K. With increasing $C^*$ for the WM96 NGWD (F2C10–F2C30, F2C30-S) the average temperature decreases, and the range of temperature increases, where the lower tail of the PDF shows a two-peak structure comparable to UA-ICON(ECHAM). The SABER data show a Gaussian distribution and do not confirm the high temperatures of the upper tail of the PDF in the UA-ICON(NWP) simulations F2C30 and F2C30-S.

Derived from the numerical experiments, one parameter setup, F2C30-S, in Table 3 corresponding to $C^* = 30$ and an adaptation of the OGWD parameters to values as indicated for F2C30-S, provides the most reasonable prediction and therefore is used for further investigations in the time slice simulation UA-ICON(NWP) in Table 2. This version reproduces best a reasonably strong westerly stratospheric jet and an easterly mesospheric jet in the northern hemisphere and their reversed counterparts in the southern hemisphere and at the same time a reasonably low temperature in the summer mesopause region. This setup is therefore the most reasonable choice to continue investigating the middle atmosphere variability.

## 5 Northern hemisphere stratospheric and mesospheric winter variability

The process of parameter optimization (tuning), presented in Sect. 4 has been done with a clear focus on the climatological state of the MLT, where the parameter optimized simulation UA-ICON(NWP) shows a clear improvement. However, parameter optimizations otherwise potentially worsen the model performance in other regions of the atmosphere, e.g. as discussed for the warm stratopause temperature bias. In this section, we focus on the NH stratospheric and mesospheric winter variability, to evaluate the implication of the parameter optimization on this important dynamical aspect of the model. A key measure for the NH winter variability is the frequency of major SSWs, strongly related to troposphere-stratosphere coupling via the upward propagation of planetary wave and gravity wave activity (Sect. 5.2).



## 5.1   Seasonal cycle of northern hemisphere stratospheric variability

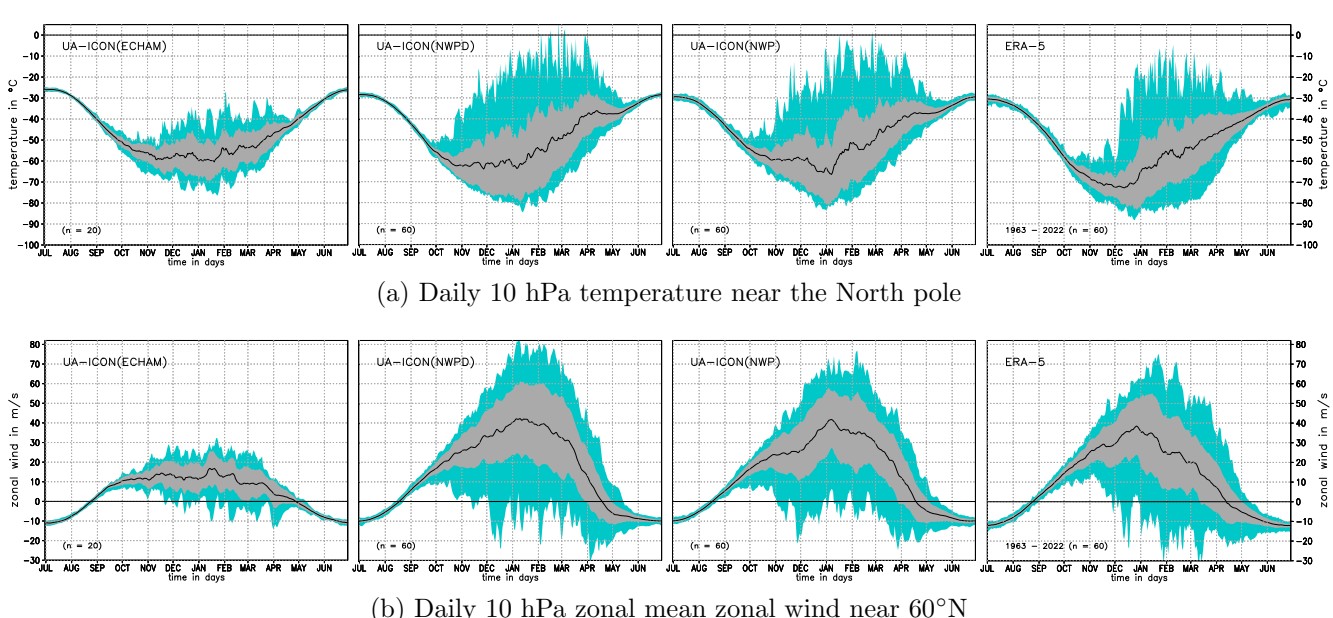

(a) Daily 10 hPa temperature near the North pole

(b) Daily 10 hPa zonal mean zonal wind near 60°N

**Figure 9.** Climatological seasonal cycle of daily 10 hPa temperature near the North pole; (b) zonal mean zonal wind near 60°N for UA-ICON simulations and ERA-5. From left to right: UA-ICON(ECHAM), UA-ICON(NWPD), UA-ICON(NWP), and ERA-5 (1963–2022). The black solid line indicates long-term averaged daily mean time series; darker shading indicates a range of ±1 standard deviation around the average; lighter shading indicates the maxima/minima reached within the complete daily data sets.

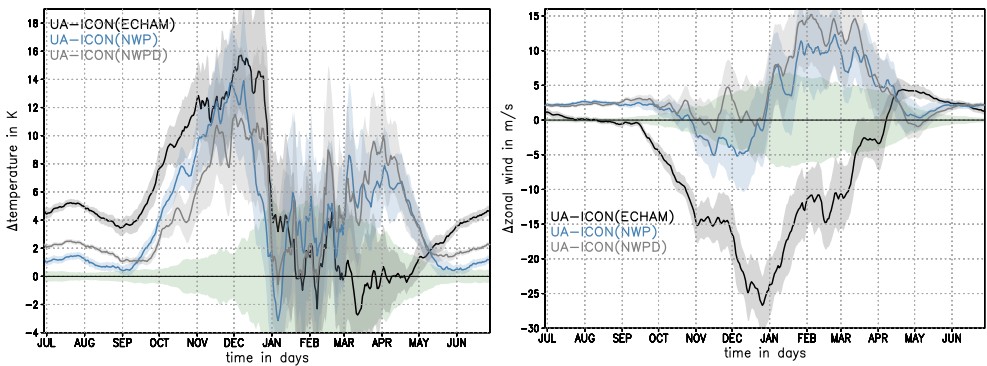

**Figure 10.** Climatological seasonal cycle of daily deviations of UA-ICON simulations from ERA-5; (left) 10 hPa temperature near the North pole; (right) 10 hPa zonal mean zonal wind near 60°N. The shading represents the 99% confidence interval of the daily mean of the individual simulations; light green shading around the zero line indicates the 99% confidence interval of the ERA-5 data.





Figure 9 shows the climatological seasonal cycle of NH stratospheric (10 hPa) polar temperature and mid-latitude (near
60°N) zonal wind for UA-ICON(ECHAM), UA-ICON(NWPD), UA-ICON(NWP), and ERA-5 (1963–2022) with the black
solid line. Overlaid with lighter shading are the daily mean maxima and minima occurring within the 60 years (20 years for
UA-ICON(ECHAM)) in each dataset. The overlaid darker shading gives the $\pm 1$ range of the standard deviation about the
daily averaged time series. Figure 10 shows the deviations of the climatological seasonal cycles presented in Fig. 9 for the
UA-ICON simulations compared to ERA-5. We start with a discussion of the daily climatological time series (black solid line
in Fig. 9) and the deviations of the UA-ICON simulations to ERA-5 (Fig. 10). The summer temperatures are too warm by
$\sim$4 K for UA-ICON(ECHAM), $\sim$1 K for UA-ICON(NWP), and by $\sim$2 K for UA-ICON(NWPD) compared to ERA-5. With
the transition to the westerly circulation in October the warm biases of both UA-ICON(ECHAM) and UA-ICON(NWP) are
getting larger, reaching a maximum of 11–16 K in December. Until mid to end of December, these positive anomalies are
outside the 99%–confidence interval of the daily ERA-5 North Pole temperature, given with the light green shading around the
zero line in Fig. 10, indicating a significant warm bias in the NH polar stratosphere. On average the warm biases for the UA-
ICON(NWP) and UA-ICON(ECHAM) simulations reduce in January and February fluctuating around 2–3 K, whereas UA-
ICON(ECHAM) in March and April shows a good agreement with ERA-5, the warm bias of the UA-ICON(NWP) increases.
The good agreement of the UA-ICON(ECHAM) simulation in 10 hPa NH polar temperatures from January to April with
ERA-5 is partly a consequence of the weak intensity of the SSWs in this simulation, as seen in the weak variability. Both
ERA-5 and the UA-ICON(NWP) simulations show intense increases in polar 10 hPa temperature during SSWs, especially
in the late winter period, contributing to an increase in the climatological mean. The 10 hPa zonal mean zonal wind near
60°N is slightly too strong for UA-ICON(NWP) simulations during the summer season, whereas UA-ICON(ECHAM) at least
from late July to the beginning of September is very close to ERA-5. However, beginning with the transition to the westerly
circulation UA-ICON(ECHAM) shows weak zonal winds with deviations to ERA-5 peaking at -25 m s$^{-1}$ in late December.
All UA-ICON(NWP) are in better agreement with ERA-5 during the NH winter season. Both UA-ICON(NWP) simulations
show in October to December moderate deviations of the zonal mean zonal wind to ERA-5, most of the time within or close to
the ERA-5 confidence interval. Later in winter from January to March, the zonal wind of UA-ICON(NWPD) exceeds the zonal
wind in ERA-5 by up to 15 m s$^{-1}$, and by 5–12 m s$^{-1}$ for UA-ICON(NWP), most of the time outside the ERA-5 confidence
interval. Concerning the winter variability, presented with the daily extremes and the range of $\pm$ one standard deviation in
Fig. 9, all UA-ICON(NWP) simulations show a far better performance than UA-ICON(ECHAM). The observational record of
the NH stratospheric winter variability, given with the ERA-5 reanalysis (Fig. 9a, b, right), indicates an increase in temperature
maxima near the North Pole in December. This is accompanied by the occurrence of easterly extremes which indicate major
SSW events. SSWs tend to be more intense and occur more often during January and February, as reflected by the warmest
maxima and the most intense easterlies in these months. The UA-ICON(ECHAM) (Fig. 9a, b, left) simulation shows a much
lower NH stratospheric winter variability, characterized by the smaller standard deviations of the wind and the temperature
difference and the less pronounced extreme values, compared to the UA-ICON(NWPD) and UA-ICON(NWP) simulations.
However, due to the weak zonal mean wind easterly winds occur and the criteria for SSWs are fulfilled frequently. In both UA-
ICON(NWP) simulations, the range of maxima/minima is comparable to ERA-5. Concerning the timing of the most extreme





temperature and zonal wind anomalies, both UA-ICON(NWP) simulations show SSW-related extremes too early in November
and too late in March, however, less easterly extremes in mid-winter, than ERA-5.

## 5.2 Major sudden stratospheric warmings

Major SSWs are accompanied by changing mesospheric and thermospheric propagation conditions for large-scale Rossby and
gravity waves, and tides. The mechanism is further illustrated in Fig. 11 with the time-height section of the averaged daily
evolution of area-weighted averaged quantities centred at day zero for SSW events detected in UA-ICON(NWP). Figure 11
(left) shows the absolute values of the quantities whereas the right column shows the respective anomalies to the long-term
daily mean. The zonal mean zonal wind (averaged from 50–70° N) (Fig. 11c, d) is decelerated in the stratosphere by Rossby
waves with westward propagating intrinsic phase speed, focusing on the polar cap as represented by the strong convergence
of the Eliassen-Palm (EP) flux $\nabla \cdot \mathbf{F}$ (averaged from 70–90° N) (Fig. 11e, f) before day zero of the SSWs exerting a strong
westward directed drag on the zonal flow. The usual eastward-directed stratospheric flow allows the upward propagation of
westward propagating NGWs and OGWs, creating a westward-directed drag in the stratosphere/lower mesosphere (OGWD)
and mesosphere (NGWD) as shown with the polar cap (70–90° N) averaged NGWD (Fig. 11g, h) and OGWD (Fig. 11i, j).
With the onset of the westward-directed flow in the stratosphere, the filter conditions for the upward propagating waves change
and the Rossby waves and westward propagating gravity waves are blocked, while eastward propagating NGWs are now in
favour of propagating upward. This is illustrated by the average time evolution of the wave drags around day zero of SSWs
in Fig. 11. A strong increase in negative $\nabla \cdot \mathbf{F}$ (convergence) starts some days before day zero in the upper stratosphere and
propagates downward. The EP-flux diagnostic $\nabla \cdot \mathbf{F}$ not only accounts for large-scale planetary waves but also includes the other
scales of the model related to resolved gravity waves, e.g. the westward-directed drag emerging in the upper stratosphere/lower
mesosphere after day zero. The westward-directed NGWD and OGWD are decreasing towards day zero and, in the case of
the NGWD, it turns eastward-directed, starting in the upper mesosphere near day zero and propagating downward near 10
hPa within 45 days. These strong changes in the wave forcing associated with SSWs have a large impact on the residual
circulation as shown by the polar cap (70–90° N) average of the meridional (Fig. 11k, l, $\overline{v^*}$) and vertical (Fig. 11m, n, $\overline{w^*}$)
components of the residual MMC. On average during NH winter conditions $\overline{v^*}$ is northward directed (positive) and strongest
in the lower mesosphere (60–70 km). In the course of SSW events, the mesospheric northward-directed $\overline{v^*}$ weakens or even
reverses to a southward-directed (negative) flow whereas the northward-directed $\overline{v^*}$ intensifies in the stratosphere. Consistently,
for continuity reasons, the average NH winter conditions of down-welling (negative $\overline{w^*}$) in the stratosphere and mesosphere
change to an upward-directed flow in the lower mesosphere and an intensification of the down-welling in the stratosphere.
The related anomalies of $\overline{v^*}$ and $\overline{w^*}$ (Fig. 11l, n) emphasize the outlined SSW-related changes in the residual MMC which are
directly related to the induced adiabatic temperature changes with strong warming in the stratosphere, a strong cooling in the
mesosphere, and again warming in the lower thermosphere (Fig. 11a, b).
For detecting major SSWs, we apply the so-called WMO criterion (Mcinturff, 1978; Labitzke, 1981). According to this,
two conditions need to be met at a pressure level of 10 hPa, reversing the climatological winter conditions in the middle
stratosphere, (I) the zonal average zonal wind at 60°N ($\overline{U}_{60N}$) has to be in a westward direction (easterly wind), and (II) the



**Figure 11.** Time-height section of the UA-ICON(NWP) daily evolution of averaged quantities in a period around SSW events (left); associated anomalies of the averaged quantities relative to the respective long-term daily mean (right). At day zero (the central day, vertical dashed line) the WMO criterion at 10 hPa for major SSWs is fulfilled for the first time.





difference in temperature between the zonal average at 60°N and the North Pole ($\overline{\Delta T}$) has to be positive within a time window of $\pm 5$ days around the central date of the SSW which is the first day when condition I is fulfilled. The detection algorithm

requires at least 20 days of westerly $\overline{U}_{60N}$ between two SSWs and preceding an SSW at least 10 days of westerlies with at least one day exceeding the threshold of 5 m s$^{-1}$. With these additional constraints, we prevent counting one SSW twice and distinguish final warmings from SSWs. Charlton and Polvani (2007) introduced the 20 days, based on the calculation of thermal damping times by Newman and Rosenfield (1997), a period that approximates two radiative time scales at 10 hPa.

There is substantial variability in the number of SSWs detected per decade (Fig. 12a), and the SSW frequency reported in

the literature is therefore dependent on the analysis period, in addition to the criterion applied for the SSW detection (Butler et al., 2015). The value of 6 SSWs per decade, frequently reported (e.g. Charlton and Polvani, 2007; Butler et al., 2017), refers to SSW detection based on the 10 hPa $\overline{U}_{60N}$ only, as introduced by Charlton and Polvani (2007). The WMO criterion applied in this study gives slightly lower SSW frequencies ($\sim$4.8–5.8 SSWs/decade) for the reanalyses, as it is more strict by additionally taking into account the $\overline{\Delta T}$ condition. Figure 12 shows the statistical evaluation of SSWs in the UA-ICON

simulations in comparison to reanalyses from NCEP/NCAR (1963–2022), ERA-5 (1963–2022), and MERRA2 (1980–2022). The SSW frequency and additional SSW statistics characterizing major SSWs on average are summarized in Tables 4 and 5 for the UA-ICON time slice simulations and the reanalyses (Zülicke et al., 2018). These are in Table 4 the SSW frequency per decade ($F_{SSW}$), the event duration in days (D), the maximum 10 hPa, 60° N easterly zonal mean zonal wind speed within an event in m s$^{-1}$ ($E_{max}$), the event accumulated easterlies in m s$^{-1}$ ($I_{acc}$).

The time series of the 10-year moving averaged SSW frequencies (Fig. 12a) exhibit large variations for the reanalyses and the UA-ICON(NWP) simulations, whereas the variations for the UA-ICON(ECHAM) simulations are relatively small. The drop to zero for the reanalyses at year 26 corresponds to the 10 years of SSW absence observed from 1988 to 1997, which incidentally is reproduced by UA-ICON(NWPD). Stratospheric variability and the variations in SSW frequency of the observational record have been attributed to several forcing mechanisms acting from above, such as solar variability (Labitzke,

1987), propagating upward from the troposphere as planetary wave variations related to, e.g. variability of SST, for example, ENSO activity (Manzini et al., 2006; Domeisen et al., 2019), variations in the Eurasian snow cover (Cohen et al., 2007; Schimanke et al., 2011), or processes of internal stratospheric variability like the Quasi-Biennial-Oscillation (QBO) (Holton and Tan, 1980). However, none of these external drivers for stratospheric variability is included in the UA-ICON simulations, and the ocean surface state is based on an average over the years 1979-2016, therefore the variations in SSW frequency are

part of the intrinsic model variability. Other studies, however, emphasize the role of the stratosphere itself in acting as a wave amplifier, as discussed in de la Cámara et al. (2019). The bar charts display the number of SSWs per decade (full coloured bars) with the error bars indicating the 95% confidence interval (CI) ($F_{SSW}$ in Table 4), derived from a bootstrapping method applied individually to the complete time series of yearly SSW frequency of each dataset and re-sampling 50,000 times, with replacement, the same number of years (the same approach as in Wu and Reichler, 2020). The open, yellow bars indicate the

average intensity of the SSWs in terms of the averaged accumulated easterly wind anomalies ($I_{acc}$ in Table 4). The frequency of SSW events during the winter season from November to March (Fig. 12a, $F_{SSW}$ in Tab. 4) for the reanalyses are in the range of 4.8 (1.4 CI) to 5.8 (1.6 CI) per decade, given by NCEP/NCAR and ERA-5, respectively, whereas MERRA2 lies in between.



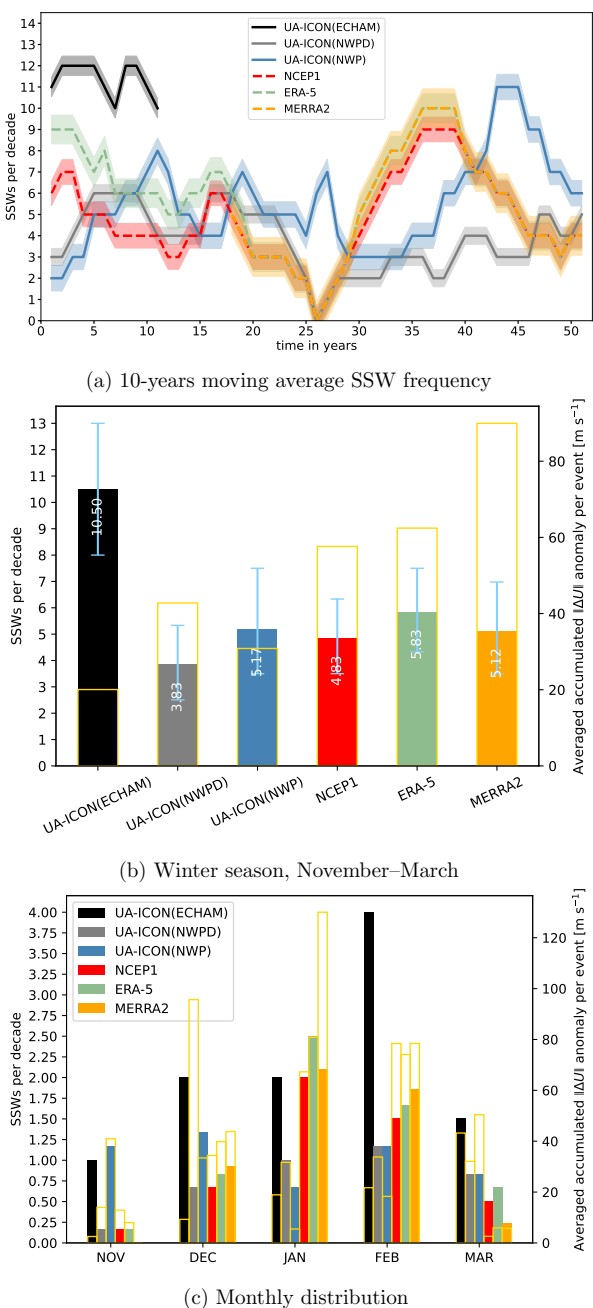

(a) 10-years moving average SSW frequency

(b) Winter season, November–March

(c) Monthly distribution

**Figure 12.** (a) Time series of the 10-year moving averaged major sudden stratospheric warming (SSW) frequency in events per decade within the NH winter season (November–March) for UA-ICON simulations and reanalyses; the shading indicates the 95% confidence interval. (b) Bar chart of SSWs per decade with error bars indicating the 95% confidence interval; (c) the monthly distribution of SSWs per decade. The statistics are based on a period of 20 years for UA-ICON(ECHAM), and 60 years for UA-ICON(NWPD), UA-ICON(NWP), NCEP1, ERA-5 (1963-2022), and 23 years for MERRA2 (1980-2022).



Although NCEP/NCAR and ERA-5 statistics are derived for the same period, they differ by one event per decade, as some events are not captured by NCEP/NCAR, probably due to the relatively low model top in the stratosphere. MERRA2 gives a lower frequency of SSWs than ERA-5, but this is caused by the shorter period used for MERRA2, and limiting ERA-5 to 1980-2022 shows the same SSW frequency as MERRA2, whereas for NCEP/NCAR the SSW frequency for 1980-2022 is again lower (4.9 SSWs/decade). There is a decrease in $F_{SSW}$ and an increase in $I_{acc}$ when changing from UA-ICON(ECHAM) to UA-ICON(NWP). The high $F_{SSW}$ in the UA-ICON(ECHAM) simulation is related to the very weak stratospheric polar vortex, leading to frequent transitions to $\overline{U}_{60N}$ easterlies and a positive $\overline{\Delta T}$ during the winter period (Fig. 9), however, the average $I_{acc}$ is only low. When applying the NWP physics package $F_{SSW}$ and $I_{acc}$ improve. The UA-ICON(NWP) simulation shows $F_{SSW}$ in the range of the reanalyses but a lower $I_{acc}$ than ERA-5, NCEP/NCAR, and MERRA2, which shows the largest $I_{acc}$ among the reanalyses, whereas the UA-ICON(NWPD) simulation shows a slightly lower $F_{SSW}$ but a larger $I_{acc}$.

The monthly distribution of the SSW frequency (Fig. 12b) from November to March shows the largest frequency in January for all reanalyses, followed by February, December, March, and November. UA-ICON(ECHAM) (black bar) has equally large SSW frequencies in December and January, and peaks in February. The UA-ICON(NWPD) simulation (grey bar), applying the default GWD parameters, has an acceptable monthly distribution of SSW frequency with a maximum in February, but frequencies too low in mid-winter and too high in March. After GWD parameter optimization in the UA-ICON(NWP) simulation (blue bar), the event frequency is too high in December, November, and March, and too low in January and February. The averaged SSW intensity ($I_{acc}$) of UA-ICON(NWP) is comparable to the reanalyses only in December but underestimated in mid-winter and overestimated in November and March.

The UA-ICON(NWP) simulation also shows lower values, compared to the reanalyses, for other wind-based statistics like the event duration (D), the maximum easterly ($E_{max}$), $I_{acc}$ (Table 4), and the fraction of intense SSW events with $I_{acc}$ exceeding 100 m s$^{-1}$ ($R_{SI}$, Table 5). Concerning these statistics, the UA-ICON(NWPD) simulation with default GWD setting agrees better with the reanalyses. The necessary tuning of the OGWD harms this aspect of the NH stratospheric winter variability, the SSW duration, maximum and accumulated easterlies of UA-ICON(NWP) are consistently lower than the reanalyses.

Table 5 summarizes additional statistical evaluations of SSW events focusing on polar vortex geometry, SSW preconditioning, and the coupling with the mesosphere. Besides $R_{SI}$, these are the number and fraction of split vortex SSW events ($N_{SV}$, $R_{SV}$), the number and fraction of SSWs with mesospheric coupling ($N_{MC}$, $R_{MC}$), and the number and fraction of SSWs with zonal wavenumber-2 (W2) preconditioning ($N_{W2}$, $R_{W2}$). The classification of the polar vortex geometry is based on the method of Charlton and Polvani (2007), applying an elliptic vortex diagnostic in 21 days around day zero. For this purpose, one ellipse is fitted to the pressure field if there is one low-pressure system or two ellipses if there are two low-pressure systems. Thereby displaced (DV) and split polar vortices (SV) are distinguished. The detection of the SSWs with mesospheric coupling is based on polar cap (60–90°N) area weighted averaged temperature anomalies ($\Delta\overline{T}_{60-90N}$) at 10 and 0.01 hPa. If the 10-hPa stratospheric warming and the mesospheric cooling both exceed one standard deviation, the SSW is a mesospheric coupling event. The criterion for W2 preconditioning is based on the method of Bancalá et al. (2012) and requires the amplitude of the 50-hPa geopotential height W2 to be larger than the respective W1 by more than 100 m and the 100-hPa W2 heat flux larger





than the respective W1 heat flux by more than 15 K m s$^{-1}$ in 10 days before the SSW around the day with the largest 10-hPa $\overline{U}_{60N}$ deceleration.

We start with comparing R$_{SV}$ in UA-ICON simulations to reanalyses and an analysis of CMIP6 models by Hall et al. (2021) who uses instead of R$_{SV}$ the ratio of SSWs with displaced and split polar vortex (DS). To better compare our measure to the published DS, we give the DS values in Table 5 and in brackets in the text. Using the reanalyses as an observational basis, the observed relative number of split-vortex events (R$_{SV}$) is in a range of 50-59% (1.0–0.7) with differences to a certain degree due to the re-analysis period. All UA-ICON simulations have lower R$_{SV}$, where UA-ICON(ECHAM), with 33% (2.0), shows the highest R$_{SV}$, the UA-ICON(NWPD) simulation showing the lowest with 13% (6.7) 14-16% (6.0–5.2), and the GWD-tuned UA-ICON(NWP) showing an increase to 23% (3.4). Hall et al. (2021) analysed the representation of stratospheric polar vortex variability and SSWs in CMIP6 models in comparison to ERA-5 and ERA-Interim and found for ERA-5 (1979–2020) a displaced versus split vortex ratio (DS) of 1.5. The ERA-5 DS from our vortex geometry analysis is 0.8 (R$_{SV}$ of 54%), indicating the detection of more split vortex SSW events with our algorithm. The multi-model mean DS of the CMIP6 models analysed in Hall et al. (2021) is 2.2, but the DS range is large from 0.9 to 9.0. The DS values of the UA-ICON simulations are in the range of the CMIP6 models with 2.0 for UA-ICON(ECHAM), 6.7 for UA-ICON(NWPD), and 3.4 for UA-ICON(NWP). The deficit of UA-ICON(NWP) in the proper simulation of planetary zonal wavenumber-2 also shows with the statistics of W2-preconditioning which give R$_{W2}$ values in the range of only 9–13% where reanalyses show a W2-preconditioning in 24% (NCEP1) or 23% (ERA-5) of the SSWs.

| Data source | T | N$_{SSW}$ | F$_{SSW}$ | D | E$_{max}$ | I$_{acc}$ |
|---|---|---|---|---|---|---|
| UA-ICON(ECHAM) | 20 | 21 | 10.50(2.50) | 6.4(1.9) | -3.6(1.3) | -20.0(11.1) |
| UA-ICON(NWPD) | 60 | 23 | 3.83(1.50) | 7.0(1.7) | -8.9(2.7) | -42.8(18.1) |
| UA-ICON(NWP) | 60 | 31 | 5.17(1.67) | 6.2(1.7) | -6.1(1.6) | -30.8(13.6) |
| ERA-5 | 60 | 35 | 5.83(1.58) | 8.6(2.5) | -10.4(2.6) | -62.4(26.7) |
| NCEP1 | 60 | 29 | 4.83(1.42) | 8.1(2.6) | -9.6(2.7) | -57.6(28.4) |
| MERRA2 | 43 | 22 | 5.12(1.74) | 11.0(4.0) | -12.6(3.5) | -89.9(40.9) |
| ERA-Interim/MLS | 11 | 6 | 5.45(2.73) | 15.3(8.3) | -16.1(6.7) | -127.3(95.8) |

**Table 4.** Major SSW statistics for data sources (column one) with length (T) in years; total number of major SSW events and their annual frequency (N$_{SSW}$, F$_{SSW}$ in events dec$^{-1}$), their mean duration (D in days), maximum and accumulated easterlies (E$_{max}$ and I$_{acc}$ in m s$^{-1}$) as in Zülicke et al. (2018).





| Data source | $N_{SI}$ | $R_{SI}$ | $N_{SV}$ | $R_{SV}$ | DS | $N_{MC}$ | $R_{MC}$ | $N_{W2}$ | $R_{W2}$ |
|---|---|---|---|---|---|---|---|---|---|
| UA-ICON(ECHAM) | 4 | 0.19(0.14) | 7 | 0.33(0.21) | 2.0 | 16 | 0.76(0.19) | 1 | 0.05(0.07) |
| UA-ICON(NWPD) | 8 | 0.35(0.20) | 3 | 0.13(0.13) | 6.7 | 20 | 0.87(0.17) | 2 | 0.09(0.11) |
| UA-ICON(NWP) | 9 | 0.29(0.15) | 7 | 0.23(0.15) | 3.4 | 29 | 0.94(0.11) | 4 | 0.13(0.11) |
| ERA-5 | 14 | 0.40(0.16) | 19 | 0.54(0.17) | 0.8 | – | – | 8 | 0.23(0.14) |
| NCEP1 | 11 | 0.38(0.17) | 17 | 0.59(0.17) | 0.7 | – | – | 7 | 0.24(0.16) |
| MERRA2 | 12 | 0.55(0.20) | 11 | 0.50(0.20) | 1.0 | – | – | 4 | 0.18(0.14) |
| ERA-Interim/MLS | 3 | 0.50(0.33) | 3 | 0.50(0.33) | 1.0 | 5 | 0.83(0.42) | 1 | 0.17(0.17) |

**Table 5.** Major SSW statistics for data sources (column one) with number and fraction of intense SSWs ($N_{SI}$, $R_{SI}$ in #SI/#SSW), split vortex SSWs ($N_{SV}$, $R_{SV}$ in #SV/#SSW), and SSW events with mesospheric coupling ($N_{MC}$, $R_{MC}$ in #MC/#SSW) as in Zülicke et al. (2018), the ratio of SSWs with displaced and split polar vortex (DS), and the number and fraction of SSWs with wave-2 preconditioning ($N_{W2}$, $R_{W2}$ in #W2/#SSW).

As discussed for the averaged SSW-related quantities of UA-ICON(NWP) in Fig. 11, the intense warming of the polar cap
stratosphere during SSWs is the result of anomalous wave forcing, leading to adiabatic stratospheric warming, an adiabatic
cooling in the mesosphere, and adiabatic warming in the thermosphere. Figure 13 shows this stratosphere-mesosphere coupling
with profiles of the area-weighted averaged polar cap $\Delta\overline{T}_{60-90N}$ anomalies for a time-average over 21 days centred at the onset
of individual SSWs (thin dashed lines, in blue SSWs with MC, in black without MC). The thick lines represent the averages
over the individual SSWs in blue for SSWs with MC and black without MC. The red profiles represent the polar cap $\Delta\overline{T}_{60-90N}$
anomalies for a time average over the NH winter period from November 1 to March 31, including all years. The shaded region
around the red and the blue averaged profiles represents the 95% confidence interval from a bootstrap method. The top row
of Fig. 13(a-e) shows the anomalies of the respective profiles to the long-term daily climatology with both UA-ICON(NWP)
simulations during SSWs showing more intense warm stratospheric anomalies and more intense mesospheric cold anomalies
than UA-ICON(ECHAM). The average intensity of the positive stratospheric $\Delta\overline{T}_{60-90N}$ in UA-ICON(NWP) ($\sim$9 K) is slightly
lower than Aura-MLS ($\sim$11 K). The Aura-MLS stratospheric and mesospheric $\Delta\overline{T}_{60-90N}$ is slightly more intense for the
average over MC SSWs (blue profile) which is reproduced by the UA-ICON simulation. The magnitude of mesospheric cooling
associated with SSWs is well represented in the GWD-tuned UA-ICON(NWP) simulation and compares well to Aura-MLS.
The bottom row of Fig. 13(f-j) shows the correlation of the respective $\overline{T}_{60-90N}$ at a pressure level of 10 hPa with all other
pressure levels of the UA-ICON simulations and Aura-MLS. Throughout the NH winter period (red profile), the stratosphere
is well coupled to the mesosphere, indicated by the typical anti-correlation between the stratosphere and mesosphere. The
average profile of the UA-ICON(NWPD) simulation shows the maximum anti-correlation ($\sim$-0.75) during SSWs extending
over a more extended altitude region ($\sim$70–95 km) compared to UA-ICON(ECHAM) and UA-ICON(NWP), showing the
largest anti-correlation vertically more focused near $\sim$75 km. By tuning the GWD parameterizations in UA-ICON(NWP)





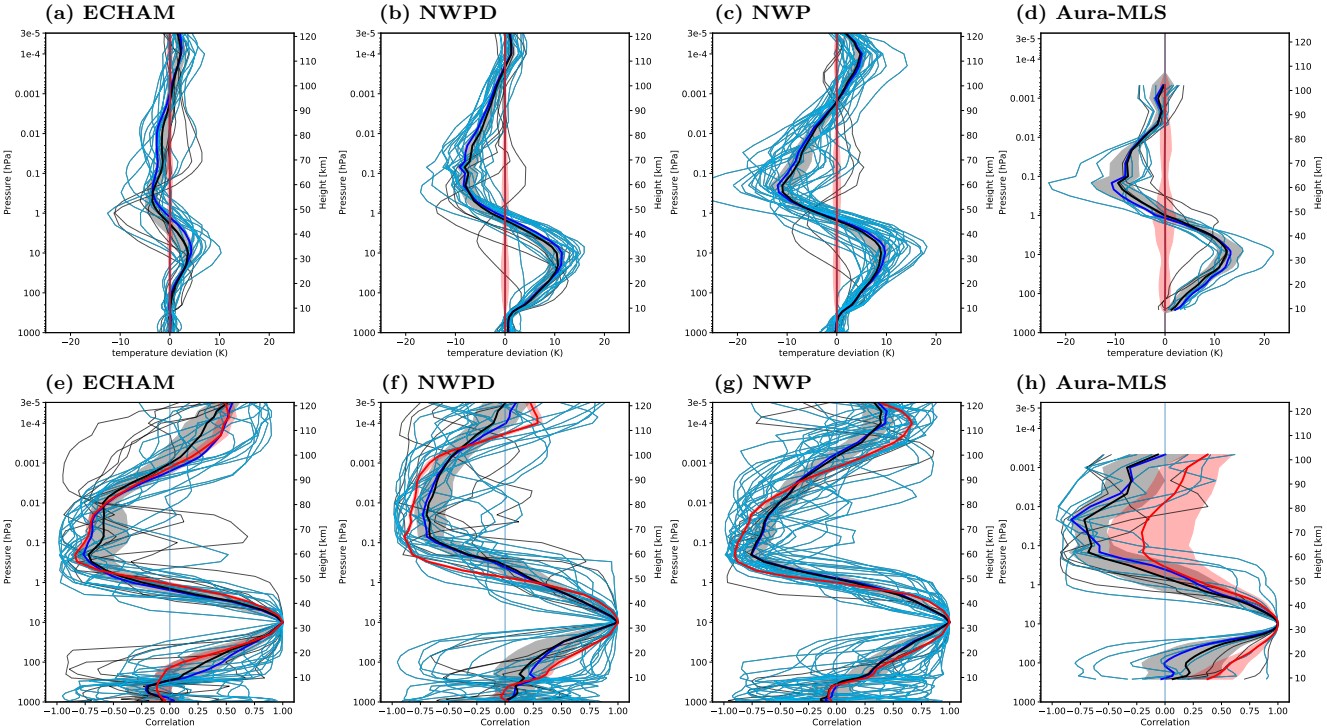

**Figure 13.** Top row: Vertical profiles of polar cap temperature anomalies ($\Delta\overline{T}_{60-90N}$) in 21-day windows centred at day 0 of SSWs (black and blue) or for the NH winter season from November 1 to March 31 (red) (a-e). Bottom row: Vertical profiles of correlations of $\Delta\overline{T}_{60-90N}$ in periods as above at different pressure levels with $\Delta\overline{T}_{60-90N}$ in the same period near 10 hPa (f-j). The profiles of individual SSWs are displayed with thin blue/black lines. The thick blue profile is the average of the individual SSW profiles with mesospheric coupling. The thick black line represents the average over all SSWs. The thick red profile represents the average of profiles for the complete winter season. The shaded regions give the 95% confidence intervals, estimated with a bootstrap method. The temperature anomalies are relative to the daily long-term mean. For UA-ICON simulations, (a,f) with ECHAM physics, (b,g) NWP physics default, (c,h) NWP physics and tuned GWD, (d,i) NWP physics and NWGD tuning, and (e,j) Aura-MLS.

the average profile based on the entire period (thick red), in general, shows a similar behaviour with the mesospheric anti-correlation focused at a lower altitude. While the vertical structure of the SSW-related anomalies in the middle atmosphere is well simulated, we find the frequency of mesospheric coupling (diagnosed with $R_{MC}$ in Table 5) slightly too high when comparing UA-ICON(NWP) with ERA-Interim/MLS. However, it is well within the uncertainties of respective data sets and thus no matter of concern.





## 6   Tidal analyses

Figure 14 presents the amplitude of the migrating diurnal (upper eight panels) and semidiurnal (lower eight panels) tides in temperature derived from UA-ICON(NWP) and SABER as a function of latitude (50°S–50°N) and height (70–110 km). The retrieval of tides uses the least-squares technique described in Yamazaki and Siddiqui (2024). The SABER tidal climatologies are based on temperature measurements during 22 years (2002–2023), while the UA-ICON(NWP) results are based on hourly temperature outputs of 60 years.

The latitude-altitude structure and seasonal variation of the migrating diurnal tide are well reproduced by UA-ICON(NWP). The latitude structure with the maximum temperature perturbation at the equator and secondary peaks at $\pm30°$ corresponds to the (1,1) Hough mode of classical tidal theory (Forbes, 1995). UA-ICON(NWP) also captures the semiannual variation in the amplitude of the migrating diurnal tide. That is, the amplitude is greater during the equinoxes than during the solstices. The semiannual variation of the diurnal tide is well known (Burrage et al., 1995) and is generally attributed to the change in the

background atmosphere, which affects the vertical propagation of the tide (McLandress, 2002a, b).

Compared to the migrating diurnal tide, the migrating semidiurnal tide has a longer vertical wavelength and thus can propagate deeper into the thermosphere (Forbes, 1995). UA-ICON(NWP) reproduces a rapid increase of the semidiurnal tidal amplitude above ∼95 km. The model also reproduces the seasonal variation with larger amplitudes during June and September than during December and March. However, the model tends to underestimate the amplitude in all seasons.

The migrating diurnal and semidiurnal tides in the MLT region from UA-ICON(NWP) also compare well with those from other numerical models such as WACCM-X (e.g., Liu et al., 2018a) and eCMAM (e.g., Beagley et al., 2000). Like UA-ICON(NWP), both these models produce a realistic seasonal variability of migrating tides in the MLT region. However, tidal amplitudes are slightly overestimated in eCMAM (e.g., Gan et al., 2014) and underestimated in WACCM-X (e.g., Liu et al., 2018b) when compared to SABER temperature observations. Tides from UA-ICON(NWP) do not differ significantly from

eCMAM and WACCM-X, implying that UA-ICON(NWP) is at least as capable as both these models in producing migrating tidal variability in the MLT region close to observations.

We also examine the tidal variability during SSWs as simulated by UA-ICON(NWP). It is well known that tides at MLT altitudes can be significantly altered during SSWs (e.g., Pedatella et al., 2014; Siddiqui et al., 2022). Especially, an enhancement of the migrating semidiurnal tide is a robust feature that has been reported for different SSWs (e.g., Jin et al., 2012; Maute et al.,

2015; Siddiqui et al., 2021). Figure 15 presents examples of the migrating semidiurnal tide response to major SSWs over four boreal winters in UA-ICON(NWP). In each winter case, the top panel depicts the zonal-mean zonal wind at 60°N and 10 hPa ($\overline{U}$), along with the difference in temperature between the North Pole and the zonal average at 60°N at 10 hPa ($\overline{\Delta T}$). Reversal of the zonal-mean zonal wind, accompanied by a reversal of the meridional temperature gradient, signifies the occurrence of a major SSW. The bottom panel shows the amplitude of the migrating semidiurnal tide in temperature at 110 km derived using

the method described in Yamazaki (2023). In all cases, the enhancement in the tidal amplitude is observed following the peak reversal of the zonal-mean zonal wind, consistent with earlier studies. The tidal response is similar during other boreal winters





that contain major SSWs, which are not presented here. These results suggest that UA-ICON(NWP) may be well suited for studying the coupling between SSWs and variability in the middle and upper atmosphere.

## 7 Discussion, summary and conclusions

This work introduces a tuned version of the upper-atmosphere extension of the ICON model with the NWP physics package (UA-ICON(NWP)). It reasonably well represents the mean state and the variability of the MLT. Here, we document the parameter optimization for the Warner and McIntyre (1996) (WM96) gravity wave parameterization for the non-orographic and the Lott and Miller (1997) (LM97) parameterization for the orographic gravity waves in UA-ICON with the NWP physics package to achieve the presented results. To this aim, we apply UA-ICON(NWP) at a horizontal resolution of R2B4 ($\sim$160 km) and 120

layers up to an altitude of 150 km. By a series of perpetual January simulations, we demonstrate the effects of changing the tunable parameters of both, the orographic and non-orographic, gravity wave parameterizations. Based on these simulations, we chose one parameter setup to best perform for the middle and upper atmosphere concerning the climatology in the MLT and the variability during the northern hemisphere winter season. We recommend for the non-orographic WM96 parameterization to increase a dimensionless factor of the saturation momentum flux density spectrum and to decrease the total launch

momentum flux and, in the LM97 parameterization, to adapt the low-level wake drag constant, the gravity wave drag constant, and the critical Froude number, as detailed in Table 3. With these settings, UA-ICON(NWP) has a sufficiently strong upper mesospheric eastward-directed zonal wind tendency (up to 140 m s$^{-1}$ d$^{-1}$) to drive a mean meridional residual circulation strong enough to create the required adiabatic cooling of the summer mesopause region. The climatological seasonal averages of the polar summer mesopause temperatures of UA-ICON(NWP) are as low as shown by SABER in austral summer ($< 150$ K)

and in boreal summer ($< 140$ K). However, the altitude and vertical extent of these low summer mesopause temperatures are slightly too low and too narrow in UA-ICON(NWP).

Introducing a stronger non-orographic gravity wave drag (NGWD) in UA-ICON(NWP) partly reverses the direction or decelerates the magnitude of the zonal mean zonal wind in the lower thermosphere from the prevailing eastward-directed flow in the stratosphere and mesosphere during the winter seasons to a westward-directed flow or a weak eastward-directed

flow in the MLT. The magnitude of these zonal wind changes in UA-ICON(NWP) agrees well with the URAP zonal mean zonal wind changes in these seasons. The vertical extent of the mesospheric polar vortex in UA-ICON(NWP), however, is limited to $\sim$80 km, which is common behaviour of GCMs or CCMs with extension to the lower thermosphere, running at a coarse horizontal resolution and thereby relying on the parameterization of the NGWD, e.g. HAMMONIA (Schmidt et al., 2006), WACCM6 (Gettelman et al., 2019), or UA-ICON (version ua-icon-1.0) (Borchert et al., 2019). Increasing the horizontal

resolution allows GCMs to resolve at least a fraction of the NGWD down to the mesoscale and at a sufficient horizontal resolution, they do not require GW parameterizations (e.g. Liu et al., 2014; Becker and Vadas, 2018, 2020; Stephan et al., 2020). The KMCM (Becker and Vadas, 2018) and HIAMCM (Becker and Vadas, 2020), both spectral models with a truncation of 240, 190 layers up to $1.5 \times 10^{-5}$ hPa (KMCM) and 260 layers up to $6 \times 10^{-9}$ hPa (HIAMCM) in the thermosphere, resolve horizontal wavelengths with $\lambda \sim$165 km and do not parameterize any GWs. Becker and Vadas (2020) discuss the height of

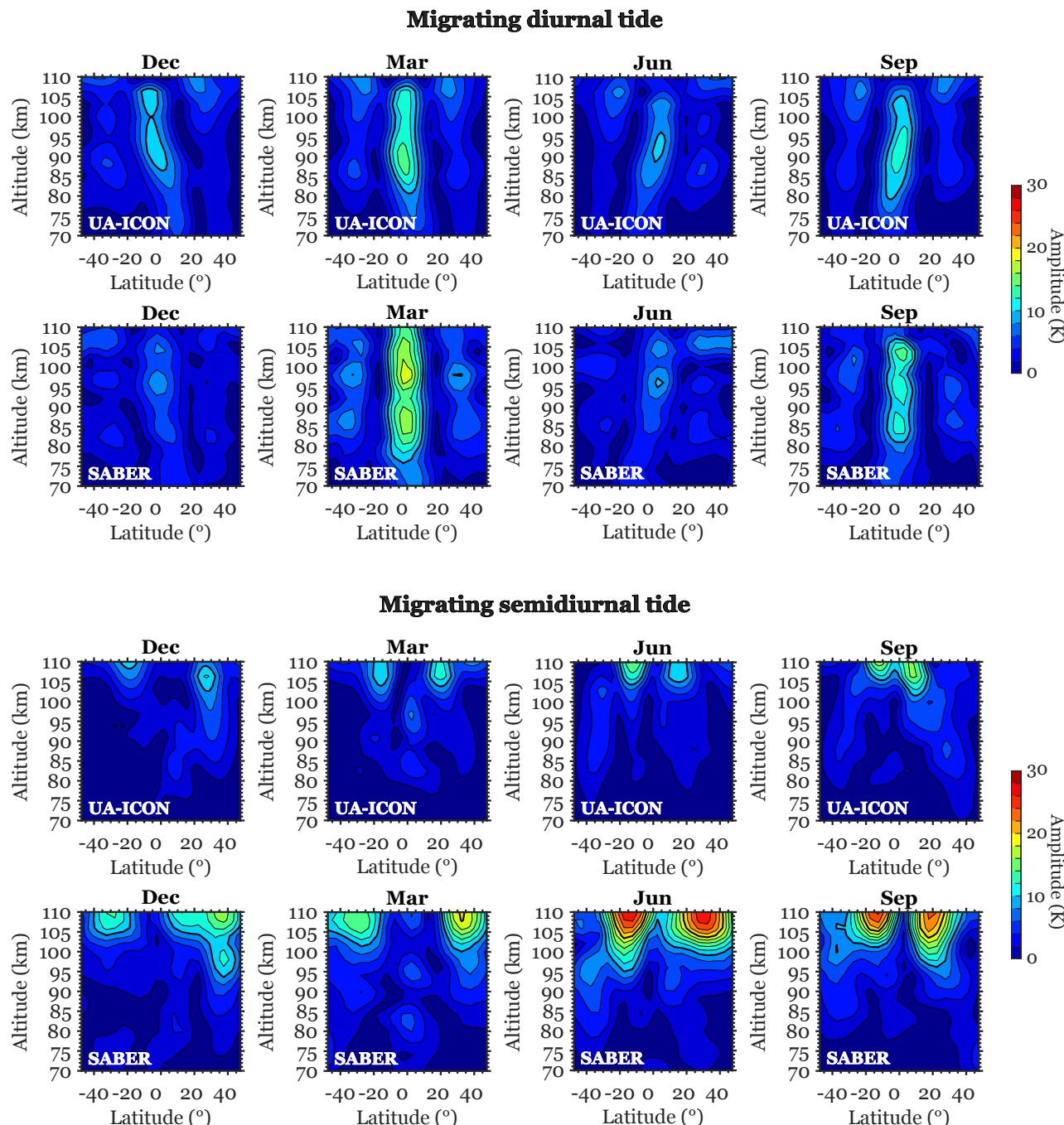

**Figure 14.** Amplitude of the migrating diurnal and semidiurnal tides in temperature as derived from UA-ICON(NWP) and SABER.

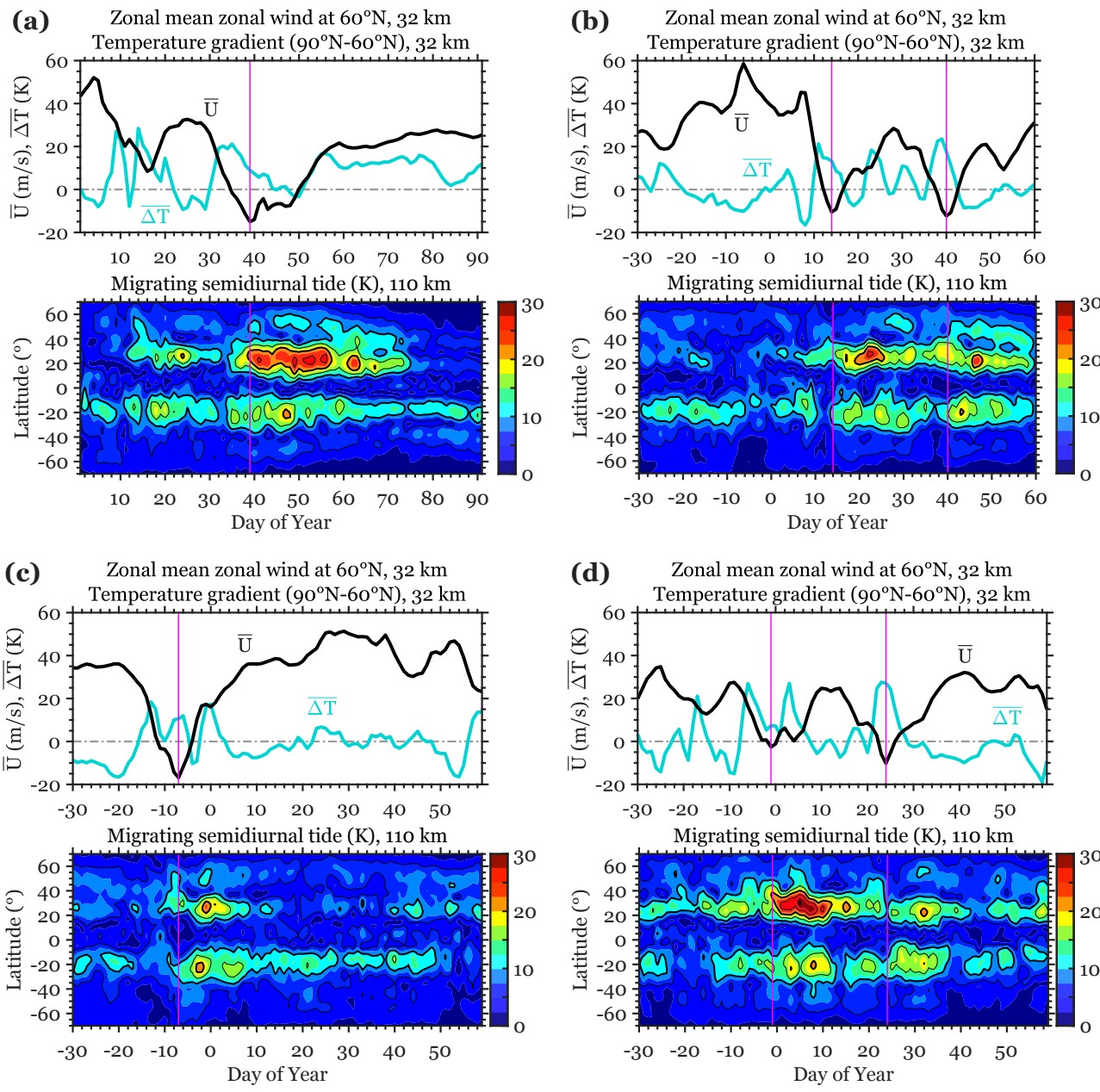

**Figure 15.** Amplitude of the migrating semidiurnal tide in temperature at 110 km as derived from UA-ICON(NWP) during four selected boreal winters containing major SSWs. The vertical lines correspond to the days of the peak reversal of the zonal-mean zonal wind.




the MLT summer zonal wind reversal in HIAMCM, which simulates GWs explicitly, and compare it to CIRA86, stating the zonal wind reversal from westward to eastward flow is too high in altitude, which is a consequence of the eastward GW drag being too high by about 10 km. This contrasts with the coarser models, relying on GW parameterizations, where the wind reversal is too low in altitude. While both types of models, GW resolving and GW parameterizing, have problems modelling the summer MLT wind reversal correctly, Becker and Vadas (2020) state that on average, GW-resolving GCMs do not simulate the MLT eastward to westward reversal in winter, which is at least in better agreement with climatologies derived for local radar-observations. Smith (2012) discusses the discrepancies between zonal mean zonal wind climatologies derived from satellite data and localized radar observations and explains the deviations with planetary-scale variations which cause persistent longitudinal variations in the zonal wind. Hindley et al. (2022) show the difference of meteor radar-derived zonal wind over South Georgia (54° S, 30° W) and WACCM6. Both show the wind reversal from the westward to the eastward zonal wind in summer however, they differ in winter with the meteor radar showing an eastward zonal wind throughout the MLT as an extension of the polar vortex into the upper mesosphere, whereas WACCM6 shows the transition from eastward to westward zonal wind. UA-ICON(NWP) behaves similarly to WACCM6 which is typical for models with parameterized GWs. Using UA-ICON (version ua-icon-1.0 with ECHAM physics) in a high-resolution configuration (R2B7 ~ 20 km horizontally, 180 layers) without any parameterized GW drag, Stephan et al. (2020) show that the model sufficiently generates resolved GW momentum flux in the MLT region to model realistic thermal and dynamic structures. Still, it remains computationally a challenge for future model applications to use wave-resolving UA-ICON versions. So far, for long simulations or ensembles, we recommend our tuned low-resolution version for an efficient simulation of the MLT region.

The mesospheric mean meridional residual circulation (MMC), driven by the dissipation of GWs, connects the summer MLT region with the respective lower mesospheric and stratospheric winter atmospheres, which results in a descending motion in the high latitudes and adiabatic warming of the stratopause regions. With increasing the parameterized GW-drag in UA-ICON(NWP) the MMC intensifies with the consequence that the winter mesopause temperatures rise, creating a warm bias compared to SABER observations. This is a drawback for the proper simulation of the summer MLT temperatures. It remains to investigate whether transient and horizontal GW propagation allowing non-orographic gravity wave parameterizations, (e.g Bölöni et al., 2021), could resolve the current numerical dilemma.

A relevant benchmark test for GCMs/CCMs extending to the mesosphere or lower thermosphere is their ability to model major SSWs with a realistic frequency and strength, including the related upward coupling to the mesosphere. Compared to the initial version of UA-ICON, mostly based on the ECHAM physics package (Borchert et al., 2019), the version presented in this work, based on the NWP physics package, has a significantly better representation of the stratospheric/mesospheric Northern Hemisphere winter polar vortex and its variability. Our statistical evaluation of SSWs includes a mesospheric coupling diagnostic (Zülicke et al., 2018), a geometric vortex diagnostic distinguishing split from displaced vortex SSW (Charlton and Polvani, 2007), and a wave preconditioning diagnostic (Bancalá et al., 2012). The overall frequency of 5.2 SSWs per decade is well in the range of 4.8–5.8 SSWs per decade, estimated from the reanalysis products. However, their intensity as quantified with the accumulated easterlies ($I_{acc}$) was found to be -31 m s$^{-1}$, less than the range of the observed -58–-90 m s$^{-1}$. Mesospheric and stratospheric temperatures are usually anti-correlated and this structure is simulated well with UA-



ICON(NWP). Although the SSW events with a strong mesospheric response are slightly too frequent, the close coupling of the mesosphere with the stratosphere is well included. Hence, the model performance in the stratosphere is essential for the model performance in the mesosphere, in a statistical sense.

The tidal analyses showed the amplitude of the migrating diurnal and semidiurnal tides in temperature to be well represented in UA-ICON(NWP), with the latitudinal structure and the seasonal variability in an acceptable state, compared to SABER-

derived tides. The enhancement of the migrating semidiurnal tide during major SSWs is well reproduced in UA-ICON(NWP), indicating a good representation of vertical coupling mechanisms in the model.

In conclusion, the UA-ICON (version ua-icon-2.1) at a horizontal resolution of ∼160 km is a highly performing upper atmosphere model available for vertical atmosphere coupling studies and investigating the MLT region. Besides, this paper has pointed out several challenges which need to be tackled in atmosphere modelling with UA-ICON. These are the lack of ability

to tune equally well stratosphere and mesosphere, the equatorward bias of jet positions, the possibly related biases of the polar vortex in SSW strengths and wave-2 preconditioning, the too narrow and low cold summer MLT, or the winter polar vortex not extending high enough. We expect to solve these problems with high-resolution modelling to increase the fraction of the resolved GW drag, a method UA-ICON is particularly designed for.

*Code availability.*   The model source code of ua-icon-2.1 used for the UA-ICON(NWP) simulations is published on Zenodo (Kunze et al.,

2024), https://doi.org/10.5281/zenodo.13927891. It is based on the ICON open source release (ICON partnership (DWD, MPI-M, DKRZ, KIT, and C2SM), 2024), ICON release 2024.01. World Data Center for Climate (WDCC) at DKRZ. https://doi.org/10.35089/WDCC/IconRelease01, which is available under a BSD 3-clause license (see https://www.icon-model.org, last access Oct 2024).

*Data availability.*   The data to reproduce the Figures will be made available on a public data server. ERA-5 reanalyses on pressure levels are available from the Copernicus Climate Data Store (Hersbach et al., 2023, https://doi.org/10.24381/cds.bd0915c6). MERRA2 reanalyses are

available from the Global Modeling and Assimilation Office (GMAO) (2015), MERRA-2 inst3_3d_asm_Np: 3d,3-Hourly,Instantaneous,Pressure-Level,Assimilation,Assimilated Meteorological Fields V5.12.4, Greenbelt, MD, USA, Goddard Earth Sciences Data and Information Services Center (GES DISC), Accessed: 2023-07-25, https://doi.org/10.5067/QBZ6MG944HW0. NCEP/NCAR reanalyses are available at the URL https://downloads.psl.noaa.gov/Datasets/ncep.reanalysis/Dailies/pressure. SABER v2.0 data are available at the URL https://data.gats-inc.com/saber/custom/Temp_O3_H2O/v2.0/. Aura-MLS data are available at the URL https://data.gesdisc.earthdata.nasa.gov/data.

*Author contributions.*   MK adapted the UA-ICON parameterisations to achieve UA-ICON (ua-icon-version-2.1). CZ ran the stratosphere-mesosphere coupling diagnosis. TS and YY conducted the tidal analysis. MK, CZ, TS, CCS, YY, CS designed the study and discussed and interpreted the results. SB and HS advised in model development and discussed results. All authors contributed to and agreed with the manuscript.



*Competing interests.* The contact author has declared that neither of the authors has any competing interests.

*Acknowledgements.* This work used resources of the Deutsches Klimarechenzentrum (DKRZ) granted by its Scientific Steering Committee (WLA) under project ID 1233.





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
