# Peer review of "UA-ICON with NWP physics package (version: ua-icon-2.1): mean state and variability of the middle atmosphere"

_Geoscientific Model Development, 2024_

## Author Comment (AC1)

**Answer to Mr García Rodríguez, Referee #1**

We thank Mr García Rodríguez for his comments and suggestions. We answer point by point in the following with the reviewer's comments added in *red/italics*.

*From a software engineering perspective applied to climate models, it would be interesting to gain a deeper understanding of the architectural decisions behind the development of the model's code. Therefore, we pose the following questions:*
*For the new configuration, were specific design patterns such as Facade or Dependency Injection used to structure the model's code? This would ensure the separation between physical parameterisations, dynamic processes, and the model logic.*

The upper atmosphere extension of the ICON model was already developed by Borchert et al. (2019) for ICON's two physical packages available at that stage. We base our adaptations on the existing model structure and design, which was created by the partners of the ICON consortium. Changing this model structure is beyond the scope of this study.
The aim of the paper is the tuning of the model performance in the middle atmosphere. We focused on gravity wave drag (GWD) because the orographic waves are usually taken to cure the "cold bias" in the winter stratosphere and non-orographic gravity waves are the main forcing of the dynamics in the mesosphere and lower thermosphere. Systematic variations of GWD parameters were undertaken (see Tab. 3), and their success was evaluated by analysing the typical wind and temperature structures in the stratosphere and mesosphere. This does not exclude the additional influence of other factors, such as prescribed SST and ozone on planetary wave structures and some warm bias in the lower stratosphere. However, the goal of qualifying UA-ICON for middle atmosphere applications of global circulation and stratospheric warming anomalies was achieved.

*To coordinate the interactions between the different physical and dynamic parameterisations, do you use patterns such as Observer or Mediator? If so, what benefits have you observed in terms of performance, maintainability, or scalability?*

None of these tools have been used.

*To manage complex parameterisations (such as orographic gravity waves), have specific techniques based on patterns like Strategy been implemented?*

To enable the testing of multiple parameter sets, we have implemented the possibility to run the model in perpetual month mode. This allows for much shorter model simulations for the evaluation of the effects of parameter changes on the climatology of the model.

*To ensure the reliability of both the simulations and the implemented code, have automated testing frameworks or static code analysis tools, such as pFUnit and FortranAnalyser, been employed in the development process? If so, how have they contributed to identifying and addressing potential issues in the codebase? The use of tools such as FortranAnalyser would be interesting to mention in order to be able to verify that the quality of the developed code is maintained or improved with the development of new versions of the software.*

Tools for code analysis, such as the mentioned FortranAnalyser, have not been used during the work for this manuscript. However, we have checked the ICON code published on Zenodo with FortranAnalyser and got a final score for the ICON/src directory of 4.1.

**References**

Borchert, S., Zhou, G., Baldauf, M., Schmidt, H., Zängl, G., and Reinert, D.: The upper-atmosphere extension of the ICON general circulation
45    model (version: Ua-icon-1.0), Geosci. Model Dev., 12, https://doi.org/10.5194/gmd-12-3541-2019, 2019.

---

## Author Comment (AC2)

**Answer to anonymous Reviewer #2**

We thank the reviewer for her/his comments and suggestions. We answer point by point in the following with the reviewer's comments added in *red/italics*. Text added to the revised version of the manuscript is included here in *blue/italics*.

**General comments**

*The results of this study are in line with experiments carried by other groups, now and in years prior. As a reviewer, and a modeler myself, I appreciate these types of studies that are difficult to publish. There is really no science in them, they are an engineering exercise, but one that is necessary to document. With this preamble in mind, I find the manuscript too long and*

10 *delving into details that are not interesting to the general reader. I do think the study needs to be published, but it also needs some thinning, which I am suggesting in the detailed comments below.*

We thank the reviewer for the encouraging remark. We have shortened the revised manuscript by moving some of the Figures to a supplement.

15

*From a general point of view, one difficulty in tuning the gravity wave drag parameterization is the coupling with interactive chemistry. When comprehensive models like WACCM were developed, it was often found that the while a set of tuning parameters reproduced a desirable climatology, the timing of the SH warming (i.e., reversal of the winds) was unrealistic and ozone chemistry was adversely affected. In this sense, what I find missing is a discussion of the timing of the springtime warming*

20 *in the lower stratosphere, like Figure 10 in doi: 10.1175/2009JAS3112.1. Moreover, the Richter et al. paper is sorely missing from the present discussion; consider also discussing Sassi et al. (doi:10.1029/2003JD004434).*

As documented in the manuscript, the ozone is prescribed from a monthly, zonally averaged climatology derived from a HAMMONIA simulation, using MOZART chemistry. The effect of a self-reinforcing temperature bias due to increased ozone

25 depletion in the presence of an initial negative temperature anomaly does not occur in our simulations. However, this is an important issue and has to be considered when using UA-ICON with interactive chemistry, which is planned in the near future. We have added and briefly discussed Richter et al. (2010) and have added Sassi et al. (2004) when mentioning the influence of ENSO on the NH winter variability in the middle atmosphere in the revised manuscript.
The transition of the monthly mean zonal mean zonal wind from the wintertime westward-directed (easterly) flow to the sum-

30 mertime eastward-directed (westerly) flow in the SH is shown in a supplementary Fig. S4. All data of the UA-ICON simulations is interpolated to constant pressure levels for this analysis.

We have added the following sentences to the discussion section:

35 *Whereas our tuning attempt for the NGWD focused on changing the saturation conditions for NGWs and the source strength, by varying parameters without latitudinal variation, Richter et al. (2010) tuned the NGWD in WACCM3 by introducing a source-oriented GW parametrization, accounting for variability in convective activity and frontal systems. One detail of their tuning success concerns the SH springtime transition of the zonal wind, which shifts to an earlier date. The transition to an easterly flow in $60\,°\,S$ at 10-hPa occurs in early November for ERA-5 reanalysis data, which the UA-ICON(ECHAM) simulation*

40 *matches perfectly. The UA-ICON(NWP) simulations show the transition in late November, which is only slightly shifted to a later date by the GWD tuning (Fig. 4, supplement), in comparison with the WACCM experiments shown in (Fig.10 of Richter et al., 2010) which appear at end of December.*

**Detailed comments**

*Abstract, line 13: This concept is repeated many time throughout the paper and I might have missed some important details. My reservation about this statement is that the forcing (SST, etc.) is climatological: how do you expect to represent realistically the occurrence of phenomena that are so intimately associated with tropospheric weather, such as wave-2 forcing?*

Despite the fact of constant climatological forcing in the troposphere, UA-ICON shows considerable variability in all SSW statistics, including the wave-2 preconditioning. However, it is rare, but not completely absent. We find it still worth mentioning, that the model with a constant forcing has a dynamically self-generated bias in some aspects.

*Figure 1. Are these the results of one model simulation or of averaging (climatology) of several instances? If the latter, then you should be able to say something about the range of uncertainty; if the former, I would caution overinterpreting these results. In the following, when difference from observed climatologies are reported, the reader needs to know how significant they are.*

The results are based on only one model simulation in each panel. However, in the case of the UA-ICON (NWPD) and UA-ICON(NWP) simulations 60 years are included to characterize the mean state, which we believe is sufficient in this context. The largest uncertainty for the climatology arises from the effects of major sudden stratospheric warmings (SSWs). We show the large variability in the NH stratospheric polar vortex in terms of SSWs per decade (Fig. 12(a), initial submission; Fig. 9(a), revised submission), confirming that periods of high and low SSW frequency are included, leading to a robust climatological average. We have tested the climatological averages for statistical significance by comparing the UA-ICON seasonal averages to ERA-5 (temperature and zonal wind) and SABER (temperature). This reveals that in large areas significant differences between ERA-5/SABER and the UA-ICON simulations exist. The results of the significance test are included in the Figure 1 of the revised submission showing the seasonal mean temperatures. The results for the seasonal differences in zonal mean zonal wind to ERA-5 are not included in the Figure 2 (revised submission), as we compare to the URAP climatology, for which no individual monthly means are available. We have added the following statement about the significance of the differences to observations/reanalyses to Section 3 of the manuscript.

*However, overall, the differences in zonal mean temperature between all UA-ICON simulations and SABER are significant in most areas, as indicated by the non-hatched areas in Fig. 2. The same holds for the zonal mean wind compared to ERA-5, where all UA-ICON simulations show significant differences in most areas up to 80 km altitude (not shown).*

*Line 171. This is odd sequencing. You cannot show a result and say we will explain later the methodology: the reader needs to see first the methodology and then we can make an informed decision on the validity of the results. Otherwise, the study is reduced to a beauty contest. I suggest moving section 4 ahead of section 3 and streamline the discussion. This will also help the flow of the paper, which instead appears is interrupted.*

We followed the reviewer's suggestion and shortened Section 4 by moving the vertical profiles and the probability density functions to the supplement. Only the Figure showing the zonal-mean anomalies stays in the manuscript. However, although we can understand his/her objection, we would like to stay with the current sequence of the sections, as it allows us to introduce the motivation for tuning the GWD parameterizations first. After specifying the tuned GWDs, we show their spatial structure (Fig 3 and 4) and refer back to T and U (Fig 1 and 2 ) for comparison with the former structures. So we keep the zonal means from simulations and observations in one compact figure. The details of tuning experiments are thus removed from the main text to the supplement. Accordingly, the shortened Table 2 moved from Section 4 to Section 2.1 (Model and Setup), as well as the description of the parameters for the tuning of the gravity wave parameterizations. We now hope the reader will easily follow the content of Section 3.

*In the same spirit, it may help to reduce the number of cases that are compared: the following figures (through Figure 5) are more appropriate for a PhD thesis, as opposed to a journal paper. Honestly, there is way too much information which could be moved to a digital supplement, if it is supported by the Journal.*

90

We followed the reviewer's suggestion and moved Figures 5, 7, and 8 to an additional supplement, as it also allows to shorten the manuscript.

95 *Figure 5. Are these hemispheric averages? It would be more informative to exclude the tropics and average only poleward of 30 degrees in each hemisphere. More generally, it is hard to follow each single line in Figure 5. The vertical line plots are useful in the process of tuning a parameterization, but as they are, they become confusing, hard to read and not very useful in a scientific paper.*

100 The latitudinal range of the averages is varying as indicated on each of the sub-panels of Figure 5. The average over the complete hemispheres is only applied for V*. In order to make this section better readable we shifted Fig. 5 with detailed tuning information into the supplement.

*Moreover, since the arguments are clearly developed referring to a global circulation, the reader's mind goes to the zonal mean plots. It would be much more useful to describe the un–tuned simulations in the previous section and introduce the tuned GWD schemes in this section.*

To follow the reviewer's suggestion would require to split Figures 1 and 2 and has the potential to enlarge the manuscript even more, as the discussion of both types of simulations at once allows for a more compact presentation.

110

*Figure 7. I suggest marking the ERA-5 and SABER PDF with thick bold lines. This is a very busy figure; it will be hard to read for many people: I cannot tell the light green from the magenta. I recommend reducing the number of cases reported in these panels (same for Figure 8) to make reading easier. An additional figure with all the cases can be added as a supplemental image.*

115 The Figures 7 and 8 are now in the supplement, however, it is technically not possible to emphasize the ERA-5 and SABER PDFs with matplotlib.hist(histtype='step').

*Figure 8. As noted above, these figures are too busy. Also, I recommend discussing the lower stratospheric temperature, important for ozone depletion.*

120

As the Figure 8 is now included only in the supplement, therefore we have not added an extra discussion of the lower stratospheric temperature.

*Figure 10. Very difficult to tell colors apart; use different type of line style.*

125

We changed the line style for the UA-ICON(NWP) simulation to dashed, to improve comparability.

*Line 320. Aren't these biases the result of excessive dynamical heating?*

130 Dynamical heating may contribute to these biases in the stratosphere. For example, the larger temperature anomaly of the tuned UA-ICON(NWP) simulation in early winter may be related to the stronger orographic GWD. However, we see a larger influence of dynamical heating on the temperature near the stratopause by increasing non-orographic GWD.

We have added the following to the manuscript:

*The warm bias of the tuned UA-ICON(NWP) simulation is slightly larger in early winter. This larger dynamic heating is probably due to a stronger orographic GWD.*

*Fig. 11e,f. Note the persistent divergence at higher altitudes. Why?*

The divergence of the Eliassen-Palm flux (EPF) is diagnosing the drag of the resolved waves in the model. Before and after the SSWs a persistent EPF divergence near 110 km is present. This is related to eastward propagating resolved waves in the middle atmosphere up to the lower thermosphere. During the SSWs, these eastward propagating resolved waves already dissipate in the mesosphere. We have added the following to the manuscript:

*Westward directed (negative) planetary wave drag up to 100km has been found in several model simulations (Zülicke and Becker, 2013; Limpasuvan et al., 2016; Okui et al., 2021) as a result of baroclinic/barotropic instability (Sato and Nomoto, 2015). The persistent positive $\nabla \cdot \boldsymbol{F}$ (divergence) in the lower thermosphere near 110 km before and after the SSWs is related to eastward propagating resolved waves. The reason for the appearance in a focussed layer around 110 km in UA-ICON is most likely in the particular vertical structure of incorporated WM96 NGWD parameterization.*

*Fig. 11, the V\* and W\* panels. Are these necessary? Again, this looks like more a dump from a PhD thesis; these panels do not add anything to the paper, except space and length.*

The intention of showing the residual circulation is to control the consistency of response, in particular the link between dynamic warming and temperature patterns. Especially the W\* is interesting, as the vertical velocity is a prognostic variable in the ICON model. Most GCMs/CCMs including the MLT are hydrostatic and therefore diagnose the vertical velocity. Figure 11 is included to demonstrate the ability of UA-ICON to plausibly model the physical mechanism of SSWs, which also includes a consistent alteration of the mean meridional circulation. We therefore want to stay with Figure 11 as it is, including V\* and W\*.

*Figure 12. What are the empty bars? I know it is described in the text, but you don't want the reader to go hunting for that information. Repeat in the caption.*

The missing information is now included in the caption of Figure 12.

*Figure 12: (a) Time series of the 10-year moving averaged major sudden stratospheric warming (SSW) frequency in events per decade within the NH winter season (November–March) for UA-ICON simulations and reanalyses; the shading indicates the 95% confidence interval. (b) Bar chart of SSWs per decade with error bars indicating the 95% confidence interval; (c) the monthly distribution of SSWs per decade. The statistics are based on a period of 20 years for UA-ICON(ECHAM), and 60 years for UA-ICON(NWPD), UA-ICON(NWP), NCEP1, ERA-5 (1963-2022), and 23 years for MERRA2 (1980-2022). The open bars in panels (b) and (c) indicate the averaged accumulated easterlies ($|\Delta U|$) per event.*

*Table 5. I feel these statistics are an overkill. If the goal is to evaluate the climatological behavior, one doesn't need much beyond Figure 12. If the goal is instead to accurately represent individual SSW, then the whole premise is wrong, since, as I stated above, my understanding is that the boundary forcing is climatological.*

We are convinced that it is worth including the statistics in Table 5. The reviewer is right to mention that the statistics derived from the UA-ICON simulations are not directly comparable with statistics derived from reanalyses, as only climatological conditions are prescribed and an important phenomenon as the QBO is not included. As already mentioned previously, despite this constant forcing considerable variability in all SSW statistics is present and these statistics can be used for comparing the UA-ICON simulations with different physical packages and related tuning for the gravity wave parametrizations. In addition, we benchmark here the mesospheric cooling response to stratospheric warming which is an important part for further MLT studies.

*Line 495. I understand the methodology is borrowed from the YS24 paper but a little more information is necessary here. Is the "migrating" component representative of a 60-day composite, or longer?*

The reviewer is right that the retrieval of tides from SABER data by YS24 involved a 60-day moving window. This information has been included in the revised manuscript.

*It is noted that the tidal analysis of SABER data involves a 60-day window, which might lead to an underestimation of tidal amplitudes.*

*Line 505. Need to cite Pedatella et al. 2014 (doi:10.1002/2013JA019421)*

The paragraph starting at l.505 discusses climatologies of tides and their seasonal dependence. The study by Pedatella et al. (2014), on the other hand, focuses on the response of tides to SSWs, and thus it does not fit well in this paragraph. Since the Pedatella et al. (2014) reference is already cited in the paragraph starting at l.512, which concentrates on tidal changes during SSWs, we believe no changes to the text are necessary in response to this comment.

**References**

Limpasuvan, V., Orsolini, Y. J., Chandran, A., Garcia, R. R., and Smith, A. K.: On the composite response of the MLT to major sudden strato-
spheric warming events with elevated stratopause, J. Geophys. Res.: Atmos., 121, 4518–4537, https://doi.org/10.1002/2015JD024401,
2016.

Okui, H., Sato, K., Koshin, D., and Watanabe, S.: Formation of a Mesospheric Inversion Layer and the Subsequent Elevated
Stratopause Associated With the Major Stratospheric Sudden Warming in 2018/19, J. Geophys. Res.: Atmos., 126, 1–22,
https://doi.org/10.1029/2021JD034681, 2021.

Pedatella, N. M., Fuller-Rowell, T., Wang, H., Jin, H., Miyoshi, Y., Fujiwara, H., Shinagawa, H., Liu, H.-L., Sassi, F., Schmidt, H., Matthias,
V., and Goncharenko, L.: The neutral dynamics during the 2009 sudden stratosphere warming simulated by different whole atmosphere
models, Journal of Geophysical Research: Space Physics, 119, 1306–1324, https://doi.org/https://doi.org/10.1002/2013JA019421, 2014.

Richter, J. H., Sassi, F., and Garcia, R. R.: Toward a physically based gravity wave source parameterization in a general circulation model, J.
Atmos. Sci., 67, 136–156, https://doi.org/10.1175/2009JAS3112.1, 2010.

Sassi, F., Kinnison, D., Boville, B. A., Garcia, R. R., and Roble, R.: Effect of El Niño–Southern Oscillation on the dynamical, thermal, and
chemical structure of the middle atmosphere, J. Geophys. Res.: Atmos., 109, https://doi.org/10.1029/2003JD004434, 2004.

Sato, K. and Nomoto, M.: Gravity Wave–Induced Anomalous Potential Vorticity Gradient Generating Planetary Waves in the Winter Meso-
sphere, J. Atmos. Sci., 72, 3609–3624, https://doi.org/10.1175/JAS-D-15-0046.1, 2015.

Zülicke, C. and Becker, E.: The structure of the mesosphere during sudden stratospheric warmings in a global circulation model, J. Geophys.
Res.: Atmos., 118, 2255–2271, https://doi.org/10.1002/jgrd.50219, 2013.

---

## Author Response (AR2)

**Answer to Mr García Rodríguez, Referee #1**

We thank Mr García-Rodríguez for reviewing the paper again. We answer point by point in the following with the reviewer's comments added in *red/italics*. Text added to the revised version of the manuscript is included here in *blue/italics*.

5

*Thank you so much for the time taken to respond to my comments. However, I would like to address a few points in your response.*
*I understand that the model structure was inherited from the ICON model and that changing it was beyond the scope of this study. However, it is worth noting that GMD is a journal that emphasizes model development. Given that this article is cate-*

10 *gorized by you as a "Model Evaluation paper", it would be appropriate to include some discussion on it about computational aspects of the model being evaluated.*

*The guidelines clearly expose that:*
*"Model development papers in particular often include a large proportion of evaluation. Typically, this comprises a compari-*

15 *son of the performance of different model configurations or parameterizations."*
*"It is, however, common for pure evaluation papers to contain substantial conclusions about geoscience rather than about models, and such papers are not suitable for submission to GMD."*

*Considering these criteria, the current version of the manuscript has no discussion of the evaluation of the model. So, a more*

20 *detailed discussion of these aspects would be more aligned with the journal's scope. Additionally, evaluate the ICON source code with FortranAnalyser and obtain a final score of "4.1" is a relatively high score compared to the existing climate models. Highlighting this result would help to contribute to the evaluation model and underline the importance of the quality code development. It would be beneficial to mention this explicitly in your manuscript.*

25 The paper is not only a model evaluation paper in the strict sense given by the GMD guidelines cited by Mr García-Rodríguez. We have mainly focused on the tuning of the gravity-wave parameterizations, and have evaluated the result of this in a climatological sense, but also regarding stratospheric winter variability.
A model evaluation in a computational sense, as requested by Mr García-Rodríguez, to achieve robust conclusions about, e.g. scalability, would require additional simulations with different MPI configurations. This would introduce a completely different

30 topic to the manuscript and would lead to a further enlargement.
We have added the following to Section 2.1 of the manuscript:

*The model code used in this study is published on Zenodo (Kunze et al., 2024). It is based to a large extent on Fortran. We checked the code quality with FortranAnalyser (García-Rodríguez et al., 2024) and obtained a final score of 4.1, which is a*

35 *relatively high score compared to the existing climate models (M. García-Rodríguez, personal communication, February 16, 2025).*

*Regarding the questions on software engineering practices, I acknowledge that the primary objective of your study is physical model evaluation rather than software development. However, I had expected a more detailed answer according to perfor-*

40 *mance, maintainability, and scalability. Even if they were not the main focus of your study, they remain relevant to the long-term evolution of the model.*
*Software engineering aspects may not be the primary interest, but they are nonetheless significant in the broader context of model evaluation and development. While the study does not focus on software development, including some discussion on these aspects would enhance the manuscript's completeness and adherence to the standards expected by GMD.*

45

There are ongoing efforts to increase the quality of the ICON model code, e.g. within the ICON consolidated concept (s. https://gitlab.dkrz.de/icon/icon-c). As these are ongoing works, in which none of the authors are involved, we do not want to discuss them in the current paper.

50  We have added the following to Section 2.1 of the manuscript:

*The ICON model code is largely equipped with OpenACC directives and in special configurations ICON has been deployed on GPU computing architectures (Giorgetta et al., 2022). However, the upper atmosphere extension is not ported to GPU and we performed our model simulations on the CPU architecture of the DKRZ (German Climate Computing Center) HPC system.*
55  *Using the message-passing interface with 20 cores and 2560 processors, one year of simulation requires a wall-clock time of 62 minutes.*

**References**

García-Rodríguez, M., Añel, J. A., and Rodeiro-Iglesias, J.: Assessing and improving the quality of Fortran code in scientific software: FortranAnalyser, Software Impacts, 21, 100 692, https://doi.org/10.1016/j.simpa.2024.100692, 2024.

Giorgetta, M. A., Sawyer, W., Lapillonne, X., Adamidis, P., Alexeev, D., Clément, V., Dietlicher, R., Engels, J. F., Esch, M., Franke, H., Frauen, C., Hannah, W. M., Hillman, B. R., Kornblueh, L., Marti, P., Norman, M. R., Pincus, R., Rast, S., Reinert, D., Schnur, R., Schulzweida, U., and Stevens, B.: The ICON-A model for direct QBO simulations on GPUs (version icon-cscs:baf28a514), Geoscientific Model Development, 15, 6985–7016, https://doi.org/10.5194/gmd-15-6985-2022, 2022.

Kunze, M., Zülicke, C., Siddiqui, T. A., Stephan, C. C., Yamazaki, Y., Stolle, C., Borchert, S., and Schmidt, H.: Supplementary information on - UA-ICON with NWP physics package (version: ua-icon-2.1): mean state and variability of the middle atmosphere, https://doi.org/10.5281/zenodo.13927891, 2024.